# Transient kinetic studies of the antiviral *Drosophila* Dicer-2 reveal roles of ATP in self–nonself discrimination

**Raushan K Singh[1]\*, McKenzie Jonely[2], Evan Leslie[1], Nick A Rejali[3], Rodrigo Noriega[2], Brenda L Bass[1]\***

[1]Department of Biochemistry, University of Utah, Salt Lake City, United States;
[2]Department of Chemistry, University of Utah, Salt Lake City, United States;
[3]Department of Pathology, University of Utah, Salt Lake City, United States

**Abstract** Some RIG-I-like receptors (RLRs) discriminate viral and cellular dsRNA by their termini, and *Drosophila melanogaster* Dicer-2 (dmDcr-2) differentially processes dsRNA with blunt or 2 nucleotide 3'-overhanging termini. We investigated the transient kinetic mechanism of the dmDcr-2 reaction using a rapid reaction stopped-flow technique and time-resolved fluorescence spectroscopy. Indeed, we found that ATP binding to dmDcr-2's helicase domain impacts association and dissociation kinetics of dsRNA in a termini-dependent manner, revealing termini-dependent discrimination of dsRNA on a biologically relevant time scale (seconds). ATP hydrolysis promotes transient unwinding of dsRNA termini followed by slow rewinding, and directional translocation of the enzyme to the cleavage site. Time-resolved fluorescence anisotropy reveals a nucleotide-dependent modulation in conformational fluctuations (nanoseconds) of the helicase and Platform–PAZ domains that is correlated with termini-dependent dsRNA cleavage. Our study offers a kinetic framework for comparison to other Dicers, as well as all members of the RLRs involved in innate immunity.

**\*For correspondence:**
raushan.singh@biochem.utah.edu (RKS);
bbass@biochem.utah.edu (BLB)

**Competing interests:** The authors declare that no competing interests exist.

## Introduction

The RIG-I-like receptor (RLR) family of helicases play key roles in innate immunity in both vertebrates and invertebrates (*Schuster et al., 2019*; *Fairman-Williams et al., 2010*; *Luo et al., 2013*; *Ahmad and Hur, 2015*). These helicases bind viral double-stranded RNA (dsRNA) to enable an interferon pathway in vertebrates, and antiviral RNA interference (RNAi) in invertebrates. Animals encode and express long dsRNA similar to viral dsRNA (*Whipple et al., 2015*; *Blango and Bass, 2016*), and thus, a key question is how RLRs distinguish cellular and viral dsRNA to avoid an aberrant innate immune response. The founding member of the RLR helicase family, mammalian RIG-I, recognizes di- and triphosphates on the blunt termini of viral dsRNA to distinguish this nonself species from self-transcripts, which are capped or monophosphorylated (*Goubau et al., 2014*; *Schlee, 2013*). The invertebrate RLR helicase, *Drosophila melanogaster* Dicer-2 (dmDcr-2), also distinguishes termini, but in this case blunt termini are distinguished from the 3' overhangs that are found on certain cellular dsRNAs, such as microRNAs (*Welker et al., 2011*; *Sinha et al., 2015*; *Sinha et al., 2018*).

While the helicase domains of dmDcr-2 and RIG-I are similar, other domains differ in accordance with the different mechanisms of antiviral defense. Like all metazoan Dicers, dmDcr-2 is a multidomain enzyme (*Figure 1—figure supplement 1*), with an L-shape that places the helicase and PAZ domains at the base and top, respectively, and the RNase III domains at the center (*Sinha et al., 2018*). The PAZ domain of dmDcr-2 shares structural homology with Argonautes, which show higher affinity for two nucleotide (nt) 3'-overhanging (3'ovr) dsRNA termini compared to blunt (BLT) termini (*Ma et al., 2004*; *Park et al., 2011*; *Tian et al., 2014*). By interacting with the 3'ovr terminus, the

PAZ domain allows Dicer to 'measure' the distance to the RNase III domains, thus explaining the characteristic lengths of ~20–25 base pairs for siRNAs and miRNAs (*MacRae et al., 2006*; *Zhang et al., 2004*). The PAZ and adjacent Platform domain were considered the only Dicer domains that bound dsRNA termini (*Tian et al., 2014*; *Park et al., 2011*), but a recent cryo-EM structure of dmDcr-2 showed that BLT dsRNA binds to the helicase domain in an ATP-dependent manner (*Sinha et al., 2018*).

Steady-state biochemical and cryo-EM studies reveal diverse activities for dmDcr-2, including dsRNA binding, unwinding, translocation, and cleavage (*Sinha et al., 2015*; *Sinha et al., 2018*), but how these events are coordinated and coupled with ATP binding and hydrolysis in real time is entirely unclear. Transient kinetic experiments are well suited for delineating mechanistic details on a biologically relevant time scale (second–minute) (*Williams, 1991*; *Gutfreund, 1995*; *Fisher, 2005*), and importantly, such studies have been extremely informative in deciphering the role of ATP in self– nonself discrimination by RIG-I (*Devarkar et al., 2018*). Here we used a stopped-flow system and time-resolved fluorescence spectroscopy to investigate the dmDcr-2-catalyzed reaction. We find that ATP-binding to dmDcr-2 promotes initial recognition of dsRNA termini and mediates clamping by the helicase domain. ATP hydrolysis induces local unwinding, followed by reannealing of dsRNA in conjunction with directional translocation of dmDcr2 on dsRNA until it arrives at the cleavage site. Intriguingly, our studies suggest that an ATP-dependent long-range communication between the helicase and Platform–PAZ domains modulates dsRNA cleavage and siRNA release.

## Results

### Binding kinetics of dsRNA to dmDcr-2 is termini-dependent

The equilibrium binding affinity of dmDcr-2 for dsRNA depends on molecular features at the dsRNA termini (*Sinha et al., 2015*). To identify transient intermediates of the enzyme•dsRNA interaction, and to reveal the kinetic basis of termini-dependent discrimination, we performed fluorescence-based stopped-flow experiments. Cy3-end-labeled 52 basepair dsRNAs (52-dsRNAs) with BLT or 2 nt 3'ovr termini were prepared by annealing top (sense) and bottom (antisense) strands (*Figure 1A, B*; Materials and methods). dmDcr-2 binds dsRNA termini, and one end was blocked with two deoxynucleotides and biotin, to allow binding and kinetic analysis only from the Cy3-labeled end. To simplify determination of kinetic parameters, stopped-flow experiments were performed under pseudo first-order conditions (*Gutfreund, 1995*; *Fisher, 2005*) using 10-fold excess of dmDcr-2 over dsRNA. As detailed subsequently, the initial molecular events of the dmDcr-2 catalytic cycle were kinetically well separated from dsRNA cleavage, allowing analyses without blocking substrate cleavage.

Equilibrium experiments indicate binding of dmDcr-2 to 3'ovr termini is ATP-independent, while interactions with BLT termini requires ATP (*Sinha et al., 2015*). However, transient kinetic assays revealed interactions of both termini in the absence of nucleotide (black traces), and this was monitored on a long (*Figure 1C,D*) and short (*Figure 1E,F*) time scale. The BLT•dmDcr-2 complex may be less stable without nucleotide, possibly explaining why it was undetectable in prior equilibrium assays (*Sinha et al., 2015*). The kinetics of dmDcr-2 binding to BLT dsRNA were biphasic ($t1_{1/2} = 0.25$ s, $t2_{1/2} = 49.8$ s), but monophasic for 3'ovr dsRNA ($t_{1/2} = 27.8$ s) (*Figure 1I*). Prior studies suggest dmDcr-2 has two binding sites (helicase and Platform–PAZ) for dsRNA termini (*Sinha et al., 2015*; *Sinha et al., 2018*). While equilibrium experiments indicate binding to the helicase domain is ATP-dependent (*Sinha et al., 2015*), we considered that even without nucleotide, the biphasic kinetics might reflect interactions with two binding sites. Indeed, experiments performed with dmDcr-2 lacking the helicase domain (dmDcr-2$^{\Delta Hel}$) showed monophasic binding kinetics for both BLT and 3'ovr dsRNA in the absence of nucleotide (*Figure 1G,H*). Further, the half-life of binding to 3'ovr dsRNA in the absence of nucleotide was similar for dmDcr-2 and dmDcr-2$^{\Delta Hel}$ (*Figure 1I*), consistent with the idea that 3'ovr dsRNA binds to the Platform–PAZ domain.

We further investigated termini-dependent interactions of dsRNA by measuring the fluorescence lifetime of the Cy3-end-labeled dsRNA bound to dmDcr-2 (*Figure 1—figure supplement 2*; Materials and methods). To avoid cleavage of dsRNA during the longer time scale required for time-resolved measurements, we used dmDcr-2 with a point mutation in each RNase III domain (dmDcr-2$^{RIII}$; *Figure 1—figure supplement 1*). The dmDcr-2$^{RIII}$•Cy3-BLT-dsRNA complex showed two lifetimes (1.21 ns and 2.28 ns), but dmDcr-2$^{RIII}$•Cy3-3'ovr dsRNA only one (1.28 ns) (*Figure 1—*

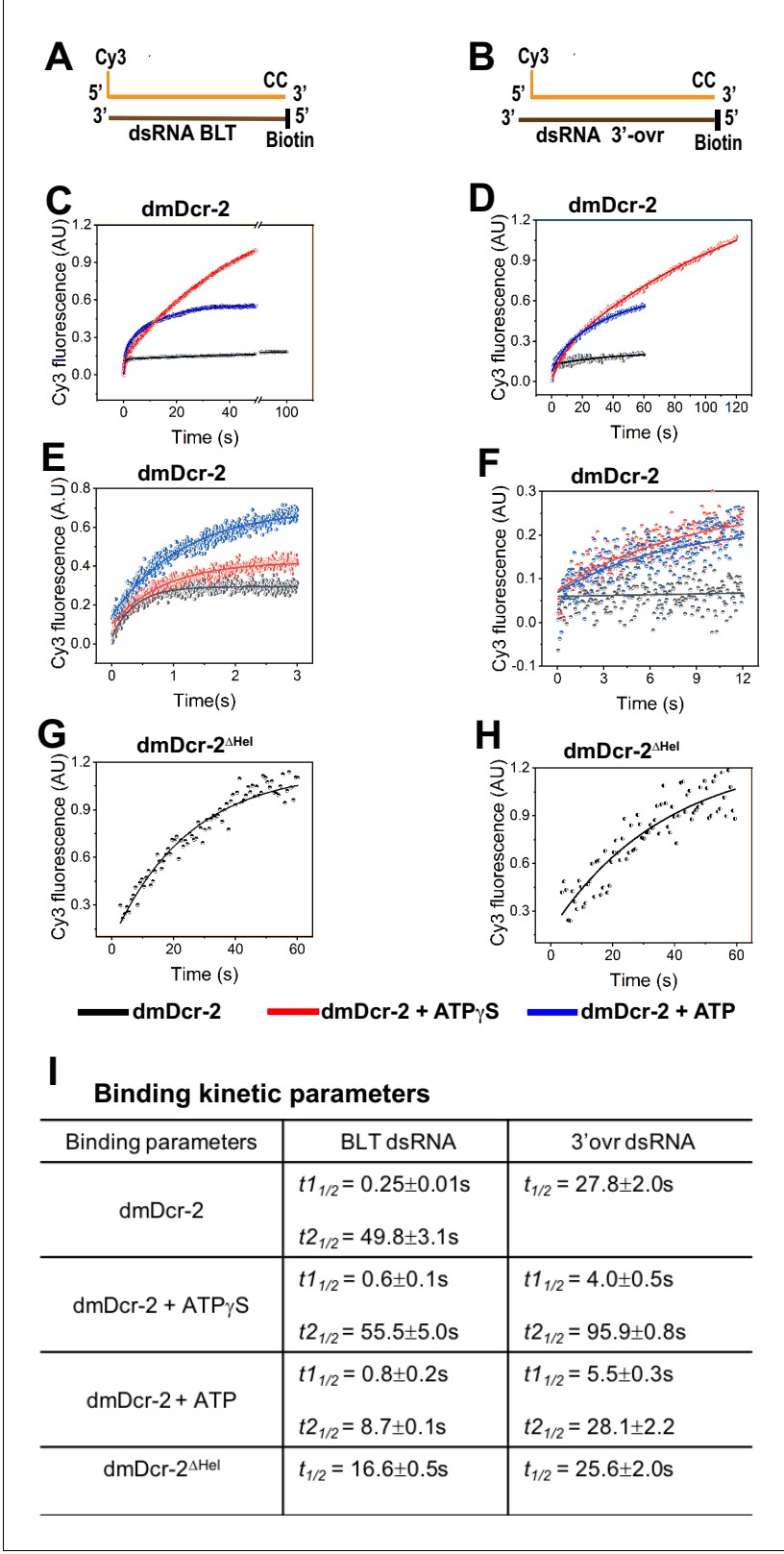

**Figure 1.** Association kinetics of BLT and 3'ovr dsRNA with dmDcr-2. The time-dependent increase in the fluorescence signal of Cy3-end-labeled 52-dsRNA (0.2 µM) was monitored upon mixing with 10-fold excess of dmDcr-2 (2 µM), alone and bound with nucleotide (ATPγS or ATP), in stopped-flow syringes. Cartoons show Cy3-end-labeled 52-dsRNA with BLT (**A**) and 2 nt 3'ovr termini (**B**) with deoxynucleotides (CC) and biotin to prevent

*Figure 1 continued on next page*

*Figure 1 continued*

dmDcr-2 binding to one end. Representative kinetic traces for binding of Cy3-labeled BLT (**C**) and 3'ovr (**D**) 52-dsRNA to dmDcr-2 are shown, with independent experiments over shorter time courses for BLT (**E**) and 3'ovr dsRNA (**F**), as well as for binding of dmDcr-2$^{\Delta Hel}$ to BLT (**G**) and 3'ovr (**H**) dsRNA. At least four to ten traces were collected for each experimental condition, and averaged trace was analyzed with single or double exponential rate equations, yielding values for kinetic parameters ($k_{obs}$ = 0.693/$t_{1/2}$) and associated standard error (**I**).

The online version of this article includes the following figure supplement(s) for figure 1:

**Figure supplement 1.** The domain organization of dmDcr-2.

**Figure supplement 2.** The fluorescence decay curve of Cy3 attached to 5'-end of BLT (**A**) and 3'ovr (**B**) 52-dsRNA while bound to dmDcr-2$^{RIII}$ (**C** and **D**) or the dmDcr-2$^{RIII}$•ATPγS complex (**E** and **F**).

---

*figure supplement 2*). Thus, in the absence of nucleotide, BLT dsRNA exists in two distinct microenvironments while bound to dmDcr-2, supporting the idea that the observed biphasic kinetics (*Figure 1C,I*) represents interactions with two sites. Moreover, the fluorescence amplitude associated with the lifetime (*Figure 1—figure supplement 2G*) suggests that a major fraction (~88%) of BLT dsRNA is bound to the helicase domain, likely because of its higher binding specificity for this domain compared to the Platform–PAZ domain (*Ma et al., 2004*). These data support the idea that in the absence of nucleotide, 3'ovr dsRNA binds to dmDcr-2's Platform–PAZ domain, while BLT dsRNA binds poorly to this domain, and primarily to the helicase domain.

In the transient kinetic experiments (*Figure 1C,D*), addition of the non-hydrolyzable analog ATPγS (red trace) or ATP (blue trace), showed biphasic kinetics for binding of dmDcr-2 to either BLT or 3'ovr 52-dsRNA (*Figure 1I*), and this was also measured over shorter times (*Figure 1E,F*). The fast-phase was similar with ATPγS or ATP, suggesting it is associated with ATP binding and unaffected by hydrolysis (*Figure 1I*). Further, the fast phase for binding BLT dsRNA ($t_{1/2}$ = 0.6–0.8 s) was approximately sixfold faster than binding to 3'ovr dsRNA ($t_{1/2}$ = 4.0–5.5 s). In contrast to the initial fast phase, the slow phase became sixfold and threefold faster, respectively, for BLT and 3'ovr dsRNA due to ATP hydrolysis (*Figure 1I*). Importantly, the half-life of this slow phase includes subsequent molecular events of the dmDcr-2 catalytic cycle, such as dsRNA unwinding/rewinding and translocation, which are coupled with the free energy of ATP hydrolysis (see below).

While the biphasic binding kinetics observed without nucleotide appeared to derive from interactions with two binding sites, ATP binding is reported to mediate a conformational change in dmDcr-2's helicase domain (*Sinha et al., 2015*; *Sinha et al., 2018*). Thus, we considered that the biphasic kinetics observed with nucleotide might reflect an initial encounter of dsRNA termini with the helicase domain, followed by a slow isomerization of the complex (dmDcr-2•ATPγS•dsRNA). We tested this possibility by measuring the fluorescence lifetime of Cy3-end-labeled 52-dsRNA bound to dmDcr2$^{RIII}$•ATPγS under equilibrium conditions (*Figure 1—figure supplement 2*). Single fluorescence lifetimes were observed, 2.15 and 1.31ns, respectively, for BLT and 3'ovr 52-dsRNA (*Figure 1—figure supplement 2E–G*), ruling out the existence of two binding sites with ATPγS, and suggesting a slow isomerization step after initial binding of dsRNA termini to dmDcr-2's helicase domain (see below). As described subsequently, this two-step binding mechanism (bimolecular followed by isomerization) was further validated by dissociation off-rate measurements, and evaluation of kinetically determined K$_d$ values using the microscopic rate constants of dsRNA interaction with dmDcr2•ATPγS. Interestingly, the initial encounter and isomerization of dmDcr-2 with BLT dsRNA are approximately sixfold and twofold faster, respectively, as compared to the corresponding steps for 3'ovr (*Figure 1I*, + ATPγS). This observation points to a kinetic control mechanism for termini-dependent discrimination of dsRNA by dmDcr-2, which could markedly impact the efficiency of an antiviral response (see Discussion). Our binding kinetics experiments revealed a two-step binding of both BLT and 3'ovr dsRNA to the helicase domain in the presence of ATPγS. Thus, the presence of ATPγS mediates a binding site switch for 3'ovr dsRNA, from the Platform–PAZ domain to the helicase domain.

## Residence time of enzyme-bound dsRNA is termini-dependent

The kinetics of dmDcr-2 binding to dsRNA with ATPγS indicates a differential mode of interaction of BLT and 3'ovr dsRNA with the helicase domain (*Figure 1I*) and is predicted to affect residence time of the enzyme-bound dsRNA. We measured dissociation off-rates for dsRNA bound to dmDcr-2 with

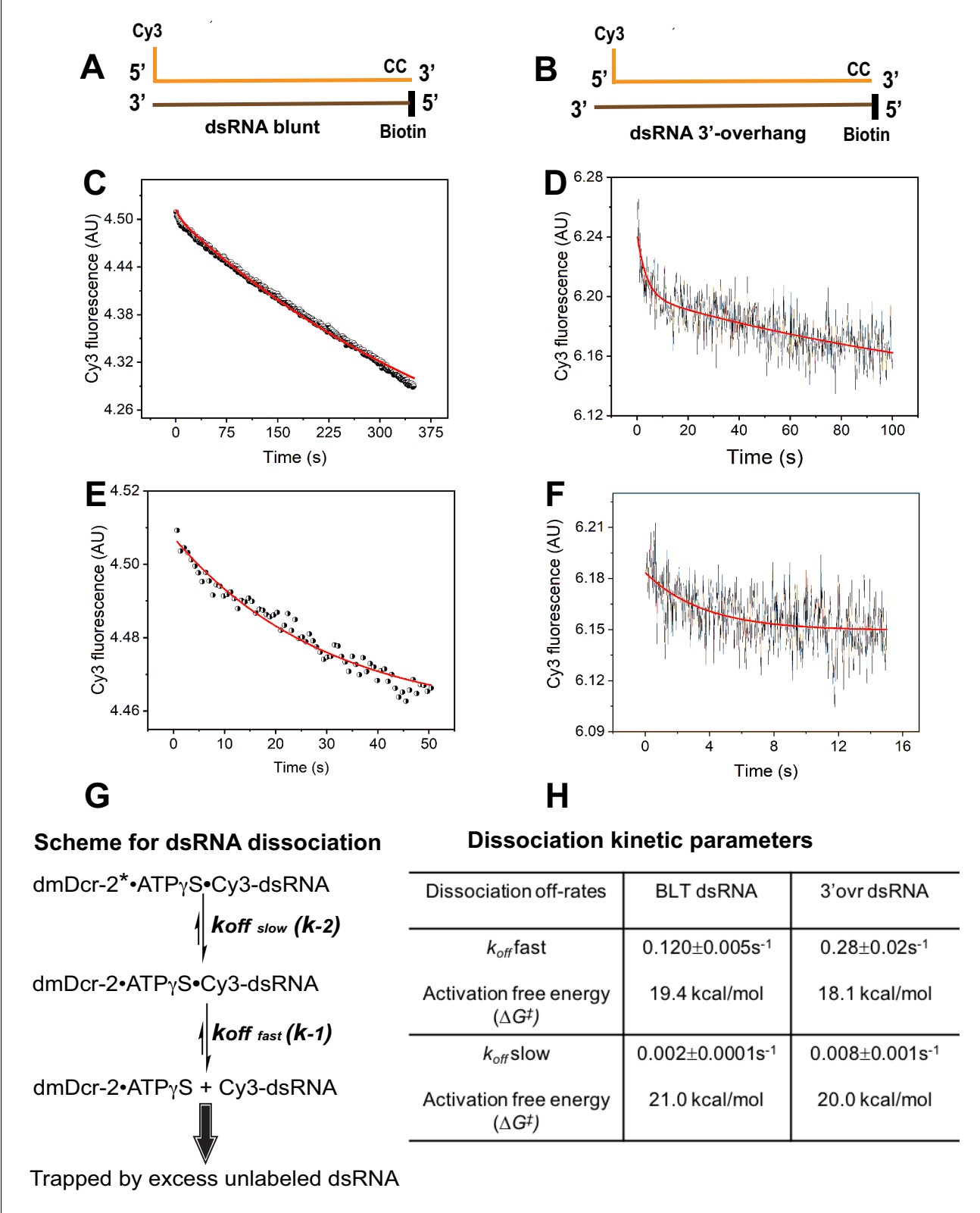

**Figure 2.** Dissociation kinetics of BLT and 3'ovr dsRNA bound to dmDcr-2•ATPγS. Dissociation kinetics were measured by mixing enzyme-bound Cy3-end-labeled 52-dsRNA with 10-fold excess of unlabeled dsRNA in stopped-flow syringes, and monitoring decrease in Cy3 fluorescence over time. Cartoons show BLT (A) and 3'ovr (B) dsRNA with representative kinetic traces for dissociation of Cy3-labeled BLT (C) and 3'ovr (D) 52-dsRNA, and independent experiments over shorter time courses (BLT, E; 3'ovr F). (G) Reaction scheme for dissociation of enzyme-bound Cy3-dsRNA; in this scheme

*Figure 2 continued on next page*

Figure 2 continued

since $k_{-1} >> k_{-2}$, the dmDcr-2•ATPγS•Cy3-dsRNA reaches steady-state level with rate constant $k_{-1}$, and then decays slowly with rate constant $k_{-2}$ (**Fersht, 1999**). (H) Kinetic traces were analyzed with single or double exponential rate equations, yielding values for dissociation off-rates. At least four to ten traces were collected for each condition, and averaged trace was analyzed with single or double exponential rate equations to obtain values for kinetic parameters ($k_{obs} = 0.693/t_{1/2}$) and associated standard error.

The online version of this article includes the following figure supplement(s) for figure 2:

**Figure supplement 1.** The schemes for dsRNA association and dissociation, and evaluation of kinetically determined $K_d$ values for interaction of dsRNA with dmDcr-2•ATPγS.

ATPγS, using the stopped-flow system (**Figure 2A–H**; see Materials and methods). The dissociation of enzyme-bound BLT and 3'-ovr dsRNA in the presence of ATPγS was biphasic (**Figure 2C,D**), as also shown over a shorter time (**Figure 2E,F**); this is consistent with two-step binding kinetics (**Figure 1**) and the principle of microscopic reversibility (**Krupka et al., 1966**; **Fisher, 2005**). While values for fast and slow $k_{off}$ for BLT dsRNA were 0.12 s$^{-1}$ and 0.002 s$^{-1}$ respectively, dissociation of 3'ovr dsRNA was twofold and fourfold faster, with fast and slow $k_{off}$ values of 0.28 s$^{-1}$ and 0.008 s$^{-1}$ (**Figure 2H**). Thus, BLT dsRNA was engaged more tightly than 3'ovr dsRNA in the dmDcr2•ATPγS complex. Activation-free energy ($\Delta G^{\ddagger}$), calculated using the Eyring equation (see Materials and methods), for dissociation of BLT dsRNA, was 1.3 and 1.0 kcal/mol higher, respectively, for fast and slow phases of dissociation, as compared to values for 3'ovr dsRNA (**Figure 2H**). This emphasizes that the molecular forces holding BLT termini in the helicase domain (with ATPγS) are stronger than those engaging the 3'ovr termini. The higher *residence time* of BLT versus 3'ovr dsRNA is consistent with prior observations indicating BLT dsRNA is cleaved processively by dmDcr-2, while 3'ovr dsRNA is cleaved distributively (**Welker et al., 2011**; **Sinha et al., 2015**; **Sinha et al., 2018**; see Discussion).

We also estimated the microscopic rate constants associated with the interaction of BLT and 3'ovr dsRNA with dmDcr2•ATPγS using observed rates constants obtained from the association and dissociation kinetics (**Figures 1** and **2**) to evaluate $K_d$ values (**Figure 2—figure supplement 1**). The kinetically determined $K_d$ values for BLT and 3'ovr dsRNA for dmDcr-2 with nucleotide are similar (within experimental error) to those determined from equilibrium binding studies (**Donelick et al., 2020**; **Jarmoskaite et al., 2020**), supporting a two-step binding and dissociation mechanism for dsRNA with dmDcr-2 (**Figure 2—figure supplement 1**).

## dmDcr-2 catalyzes an ATP-dependent transient unwinding and rewinding of dsRNA

The dmDcr-2 helicase domain is structurally homologous to mammalian RIG-I (**Jiang et al., 2011**; **Kolakofsky et al., 2012**), and whether RIG-I exhibits ATP-dependent unwinding of dsRNA is controversial (**Takahasi et al., 2008**; **Myong et al., 2009**). However, a cryo-EM structure of the dmDcr-2•ATPγS•BLT-dsRNA complex showed a single-stranded RNA within the helicase domain, and biochemical assays based on strand-displacement confirmed unwinding activity (**Sinha et al., 2018**). This raised the question of why dmDcr-2 unwinds dsRNA, since presumably it would need to rewind it for subsequent cleavage by the dsRNA-specific RNase III active sites (**Nicholson, 2014**).

To directly monitor ATP-dependent unwinding of dsRNA by dmDcr-2 in real time, and to test whether rewinding occurs, we performed stopped-flow experiments utilizing BLT and 3'ovr 52-dsRNA that contained a Cy3–Cy5 donor–acceptor Förster resonance energy transfer (FRET) pair at one terminus (**Figure 3A,B**). We anticipated that transient unwinding and rewinding of Cy3–Cy5 labeled dsRNA by dmDcr-2 would produce a sequential loss and gain in FRET signal over time. Instead of donor (Cy3) fluorescence, we specifically followed the time-dependent change in acceptor (Cy5) FRET signal to monitor transient unwinding, because the fast phase of dmDcr-2 binding to dsRNA termini with ATP leads to an enhancement in Cy3 fluorescence (**Figure 1I**, BLT, $t1_{t/2}$ = 0.8 s; 3'ovr, $t1_{t/2}$ = 5.5 s) on a time scale that could interfere with the increased Cy3 signal from a distance change between the Cy3–Cy5 FRET pair upon unwinding. While dmDcr-2 alone (black) did not produce a detectable change in the FRET signal of BLT or 3'ovr dsRNA, a biphasic change in FRET as a function of time was observed in the presence of ATP (blue trace) (**Figure 3C,D**). The half-lives for ATP-dependent unwinding of BLT and 3'ovr were 0.5 s and 1.0 s, respectively (**Figure 3E**), which correlates with the twofold difference in the rate of ATP hydrolysis by dmDcr-2 while bound to these dsRNAs (**Donelick et al., 2020**). Furthermore, the fluorescence amplitudes associated with

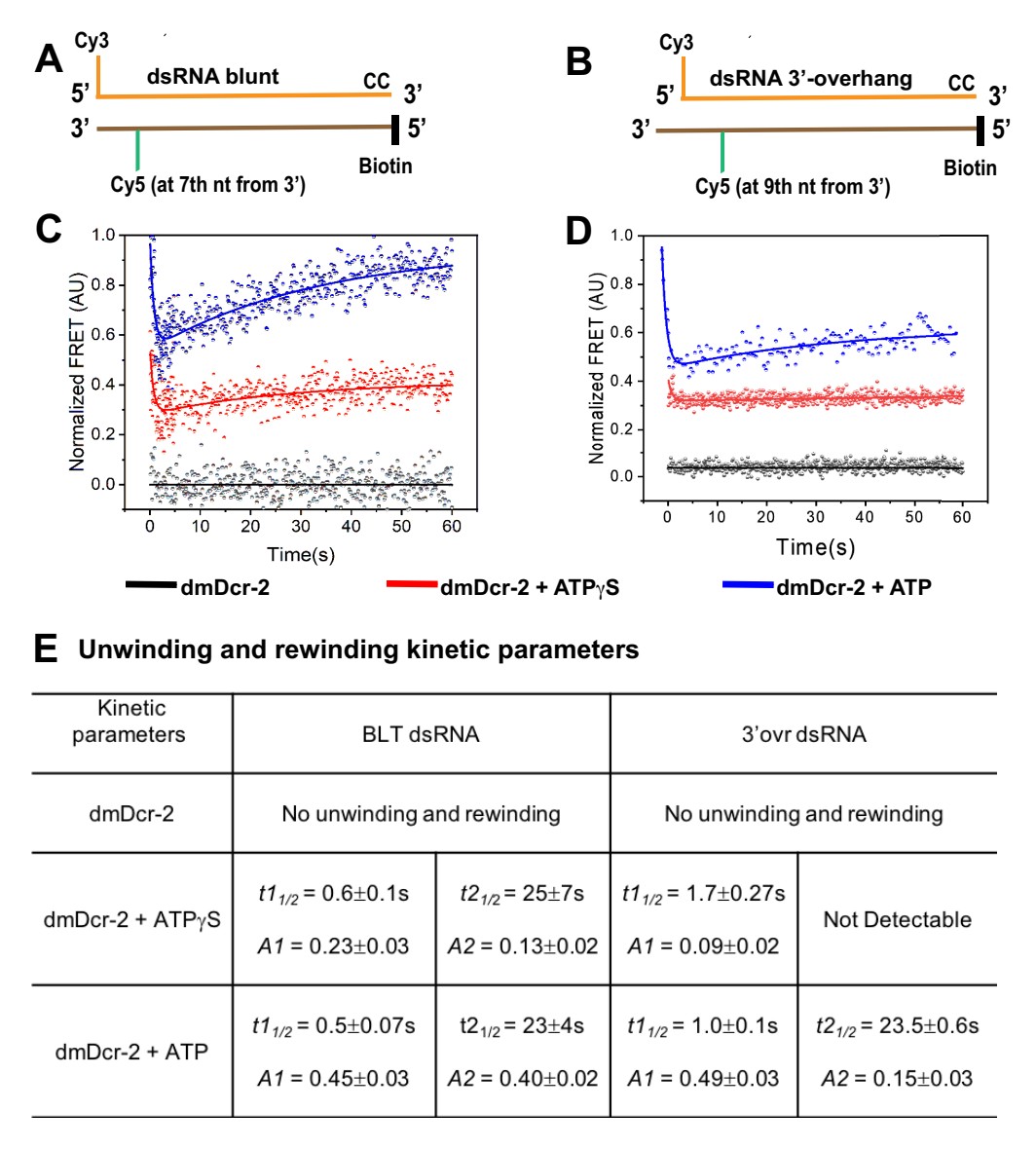

**Figure 3.** ATP-dependent transient unwinding and reannealing of dsRNA termini catalyzed by dmDcr-2. The time-dependent change in FRET signal was monitored after mixing 0.2 µM dsRNA containing a FRET (donor–acceptor) pair with 2 µM dmDcr-2 in stopped-flow syringes. Cartoons showing BLT (A) and 3'ovr (B) 52-dsRNA indicating positions of Cy3 and Cy5, and modifications to block dmDcr-2 entry, as in *Figure 1*. Representative kinetic traces for unwinding and reannealing of FRET-labeled BLT (C) and 3'ovr (D) dsRNA by dmDcr-2 alone (black), with ATPγS (red), or ATP (blue). Kinetic traces were analyzed with a double exponential rate equation, and kinetic parameters are in (E). At least four to ten traces were collected for each experimental condition, and the averaged trace was analyzed with single or double exponential rate equations, yielding values for kinetic parameters ($k_{obs} = 0.693/t_{1/2}$), amplitude (A), and associated standard error.

unwinding (*A1*) and rewinding (*A2*) in the presence of ATP were almost the same for BLT dsRNA (*Figure 3E*), that is, the FRET associated with the unwound state returned to the value observed in the initial annealed state (*Figure 3C*). By contrast, for 3'ovr dsRNA, the amplitude for the rewinding phase was only 30% of the unwinding value (*Figure 3D,E*). This suggested that, unlike BLT dsRNA, 3'ovr dsRNA remained partially unwound at the end of this assay. As observed for human Dicer, possibly the unwound state of 3'ovr dsRNA (as compared to BLT) is stabilized by interactions with the Platform–PAZ domain (*Park et al., 2011*; *Tian et al., 2014*). Such differences in the transient interaction of BLT and 3'ovr dsRNA with the Platform–PAZ domain after the unwinding/rewinding phase

likely impacts subsequent molecular events of the catalytic cycle, such as dsRNA cleavage and/or siRNA release (see below).

We also observed a sequential loss and gain of FRET signal catalyzed by dmDcr-2 on BLT and 3'ovr with ATPγS (red trace; *Figure 3C,D*), although the signal was not as robust as with ATP. This may indicate the presence of contaminating ATP in commercially purchased ATPγS, although we cannot rule out that binding of nucleotide to dmDcr-2 (without hydrolysis) is sufficient to initiate unwinding, as reported for some DEAD-box helicases (*Liu et al., 2008*). Finally, a higher fluorescence amplitude associated with the rewinding phase for both BLT and 3'ovr with ATP (compared to ATPγS) underscores the role of ATP-hydrolysis in rewinding (*Yusufzai and Kadonaga, 2008*).

## The translocation/arrival of dmDcr-2 at the cleavage site is ATP-dependent

The transient kinetic experiments described so far reveal that the initial binding, unwinding, and rewinding, are all termini-dependent, and that ATP binding and hydrolysis are critical for these early molecular events of the dmDcr-2 catalytic cycle (*Figures 1–3*). The dmDcr-2 primary cleavage site is located ~20–25 nt away from the dsRNA terminus (*Zamore et al., 2000*; *Elbashir et al., 2001*), and therefore, it is anticipated that dmDcr-2 translocates ~20 nt on dsRNA upon binding, but prior to dsRNA cleavage. We investigated the real-time arrival of dmDcr-2 at the cleavage site by monitoring the protein-induced fluorescence enhancement (PIFE) signal of Cy3 (*Hwang and Myong, 2014*; *Stennett et al., 2015*; *Nguyen et al., 2019*) attached to the 18th position of the sense strand of 52-dsRNA (*Figure 4A,B*). Representative stopped-flow kinetic traces showed the time-dependent increase in Cy3 fluorescence signal in the presence of ATP (blue trace; *Figure 4C,D*), and analyses with a single-exponential rate equation yielded half-life values for the 18-base-pair translocation of dmDcr-2 on BLT and 3'ovr dsRNA, of 14 s and 27.8 s, respectively (*Figure 4E*). The twofold higher translocation rate of dmDcr-2 on BLT dsRNA compared to 3'ovr dsRNA is consistent with the two-fold higher rate of ATP hydrolysis observed with BLT dsRNA compared to 3'ovr dsRNA (*Donelick et al., 2020*), indicating that ATP hydrolysis is directly coupled with translocation. Interestingly, the speed of dmDcr-2 translocation is comparable to that of the ATP-dependent translocation of RIG I on BLT dsRNA (*Myong et al., 2009*; *Devarkar et al., 2018*).

We also observed a time-dependent increase in Cy3 fluorescence in our translocation assay with ATPγS (red trace) or even without nucleotide (black trace) (*Figure 4C,D*), albeit rates and associated amplitudes were markedly lower compared to those in the presence of ATP (*Figure 4E*). Consistent with our binding studies (*Figure 1* and *Figure 1—figure supplement 2*), without nucleotide, 3'ovr dsRNA directly interacts with the Platform–PAZ domain, which presumably directs the dsRNA to the RNase III active sites for cleavage. Therefore, it seems likely that the enhancement in fluorescence from Cy3 at position 18 of 3'ovr dsRNA in the absence of nucleotide reflects an interaction with the protein near the RNase III domains. Similarly, with ATPγS, the observed PIFE may reflect formation of a stable, non-productive enzyme–substrate complex wherein the Cy3-label is engaged with the helicase domain upon clamping on dsRNA termini.

## ATP-dependent cleavage of dsRNA is dictated by dsRNA termini

After delineating the early kinetic events of the dmDcr-2-catalyzed reaction that precede dsRNA cleavage (*Figures 1–4*), we wished to gain mechanistic information about how these events fine-tune substrate cleavage and product release. Toward this goal, we performed transient kinetic experiments to directly monitor dsRNA cleavage and siRNA release in real time. Notably, the dmDcr-2-catalyzed cleavage of dsRNA and siRNA release was kinetically well separated from earlier molecular events (binding, unwinding/rewinding, and translocation), enabling reliable evaluation of cleavage parameters (see below; *Figures 1*, *3*, and *4*).

We prepared BLT and 3'ovr 52-dsRNA with a Cy3-Cy5 FRET pair that spanned the primary cleavage site in 52-dsRNA (*Figure 5A,B*; dotted lines), so that dmDcr-2-catalyzed cleavage coupled with siRNA release would lead to a loss of FRET. We mixed 10-fold excess dmDcr-2 (alone or with nucleotide) with FRET-labeled 52-dsRNA in the stopped-flow system and monitored the reaction for ~5 min (to observe the entire catalytic cycle) by recording the time-dependent change in Cy5 FRET signal upon excitation of Cy3 (see Materials and methods). As in the translocation assay, which used dsRNA without a FRET acceptor (*Figure 4C,D*), we anticipated that an increase in Cy3 fluorescence

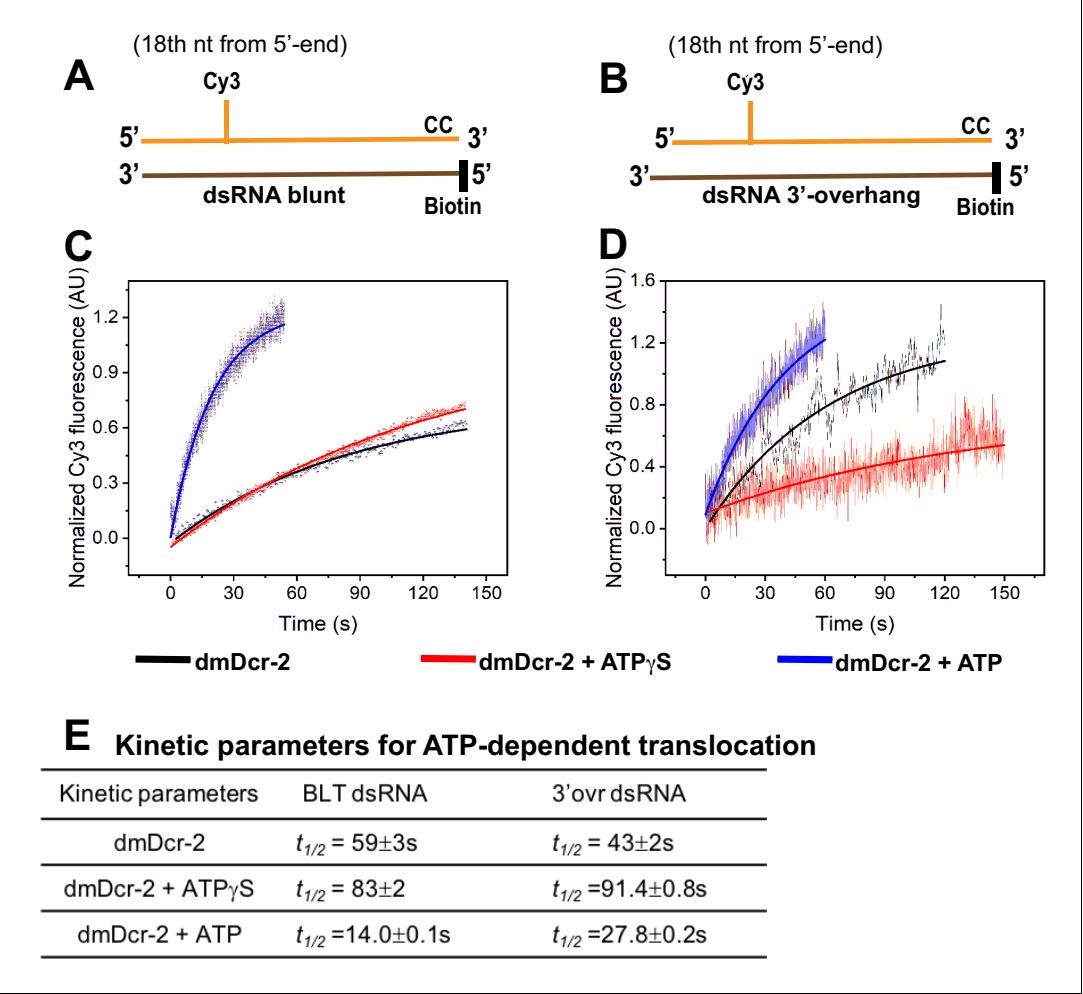

**Figure 4.** Real-time monitoring of ATP-dependent translocation/arrival of dmDcr-2 at the cleavage site. Cartoons show BLT (**A**) and 3'ovr (**B**) 52-dsRNA with a covalently linked Cy3 at the 18th nucleotide of the top (sense) strand, and deoxynucleotide (CC) and biotin to prevent binding of dmDcr-2 to one end. Representative stopped-flow kinetic traces for translocation of dmDcr-2 alone (black trace), and while bound with ATPγS (red) and ATP (blue) on BLT (**C**) and 3'ovr (**D**) dsRNA. Kinetic traces were analyzed with a single exponential rate equation, yielding observed rate constants associated with translocation (**E**). At least four to ten traces were collected for each experimental condition, and averaged trace was analyzed with single or double exponential rate equations, yielding kinetic parameters ($k_{obs}$ = 0.693/$t_{1/2}$) and associated standard error.

due to PIFE would also lead to an increased Cy5 fluorescence due to a greater extent of energy transfer from Cy3 to Cy5 (*Hwang and Myong, 2014*; *Stennett et al., 2015*; *Nguyen et al., 2019*). Indeed, the time-dependent increase in FRET signal observed for BLT and 3'-ovr dsRNA in the presence of dmDcr-2 and ATP was similar to the increase in PIFE signal observed in the translocation assay (compare $t_{1/2}$ and $t1_{1/2}$ values with ATP, *Figures 4E* and *5E*). As anticipated, the time-dependent increase in FRET signal due to arrival of dmDcr-2 at the cleavage site was followed by a slow loss of FRET for both BLT and 3'ovr dsRNA (blue, *Figure 5C,D*). We attributed this time-dependent slow loss of FRET signal to the cleavage of dsRNA coupled with siRNA release. Control experiments performed with cleavage incompetent dmDcr-2 (dmDcr2[RIII]) only showed a time-dependent increase in FRET signal (*Figure 5—figure supplement 1*), confirming this interpretation. The half-life for the ATP-dependent cleavage/product release for BLT dsRNA ($t2_{1/2}$ = 43 s) was twofold faster than that of 3'-ovr dsRNA ($t2_{1/2}$ = 83 s) (*Figure 5E*), possibly reflecting a differential interaction of dsRNA termini (after translocation) with the Platform–PAZ domain that poises the enzyme for substrate cleavage (see below) (*Sinha et al., 2015*). The twofold difference in dsRNA cleavage/siRNA release also correlates with the difference in ATP-hydrolysis rate observed with dsRNA with different termini, indicating that ATP-hydrolysis is important for substrate cleavage and/or product release (see

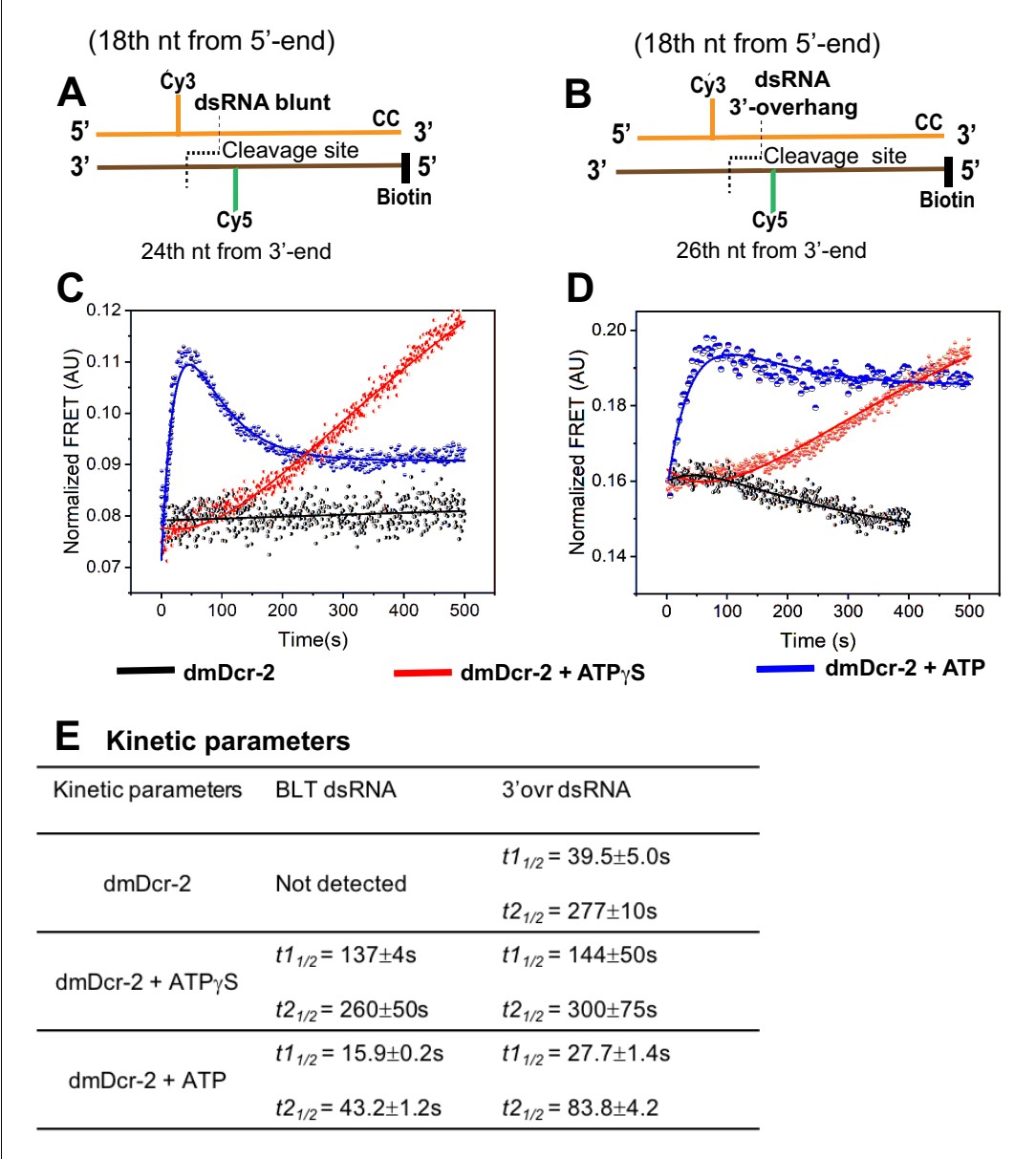

Figure 5. The arrival of dmDcr-2 at cleavage site poises the enzyme for ATP-dependent substrate cleavage and siRNA release. Cartoons illustrate BLT (A) and 3'ovr (B) dsRNA containing the Cy3-Cy5 FRET pair at positions indicated, and deoxynucleotide (CC) and biotin to block dmDcr-2 binding at one end. Representative stopped-flow kinetic traces for cleavage of BLT (C) and 3'ovr (D) 52-dsRNA by dmDcr-2 alone (black), with ATPγS (red) or ATP (blue). Kinetic traces were analyzed with a double exponential rate equation, and kinetic parameters listed in (E). At least four to ten traces were collected for each condition, and averaged trace was analyzed with single or double exponential rate equations, yielding values for kinetic parameters ($k_{obs} = 0.693/t_{1/2}$) and associated standard error.

The online version of this article includes the following figure supplement(s) for figure 5:

**Figure supplement 1.** Stopped-flow kinetics using FRET substrates designed to monitor cleavage, but with cleavage-incompetent dmDcr-2[RIII] in the presence of ATP.

**Figure supplement 2.** Stopped-flow kinetics monitoring cleavage of BLT and 3'ovr dsRNA with dmDcr2[ΔHel], and for comparison, data for dmDcr-2 alone and with ATP from *Figure 5* are shown.

below). Further, a higher amplitude for the FRET loss associated with BLT dsRNA cleavage/siRNA release compared to 3'ovr dsRNA suggests an incomplete release and/or rebinding of 3'ovr siRNA (*Gan et al., 2006*). This could arise due to a higher binding specificity of 3'ovr to the Platform–PAZ

domain, which could play a role in siRNA release or transfer to Argonaute 2 (AGO2, see below) (*Ma et al., 2004*; *Tian et al., 2014*; *Sheu-Gruttadauria and MacRae, 2017*).

In the absence of nucleotide, dmDcr-2 did not produce a significant change in FRET for BLT dsRNA, but biphasic kinetics were observed for 3'ovr dsRNA, with an initial modest gain of FRET followed by slow loss over time (black, *Figure 5C,D*), as expected for substrate binding followed by slow cleavage. This result is consistent with prior studies indicating that dmDcr-2's Platform–PAZ domain mediates a slow and ATP-independent cleavage of 3'ovr dsRNA that does not require the helicase domain (*Sinha et al., 2018*). Interestingly, we observed a similar cleavage of BLT dsRNA by dmDcr-2 lacking the helicase domain (dmDcr-2$^{\Delta Hel}$) (cyan, *Figure 5—figure supplement 2*), likely due to its binding to the Platform–PAZ domain in the absence of the helicase domain. With ATPγS, both BLT and 3'ovr dsRNA showed biphasic kinetics (red, *Figure 5C,D*) and both phases were associated with an increase in FRET, suggesting that without ATP hydrolysis, dsRNA was kinetically trapped in a non-productive conformation within the helicase domain that likely undergoes slow conformational changes.

## Conformational dynamics of helicase and Platform–PAZ domains impact dmDcr-2 catalysis

Our transient kinetics studies indicated that in the presence of ATP, BLT and 3'ovr termini are discriminated by the helicase domain during initial binding, and subsequently by the Platform–PAZ domains prior to cleavage (*Figures 1* and *5*). To gain insights into how fast conformational dynamics of these domains is correlated with catalytic activity, we performed time-resolved fluorescence anisotropy. We measured anisotropy parameters of Cy3-end-labeled BLT and 3'ovr 52-dsRNA (*Figure 6A,B*), alone (green), bound to dmDcr-2$^{RIII}$ without nucleotide (black), as well as with ATPγS (red) or ATP (blue). Our assays to this point indicated that ATPγS traps dsRNA within the helicase domain (*Figure 1* and *Figure 1—figure supplement 2*), while addition of ATP to dmDcr-2$^{RIII}$ allows translocation without cleavage, placing termini in the Platform–PAZ domain (*Figure 5*); thus, we anticipated these nucleotides would allow us to separately probe dynamics of the helicase and Platform–PAZ domains. Anisotropy decay curves (*Figure 6C–J*) were analyzed with single and double exponential rate equations yielding rotational correlation time ($\phi$), fractional amplitude ($A$), limiting anisotropy ($r_\infty$), and change in anisotropy ($\Delta r$) (*Figure 6K*; Materials and methods) (*Lakowicz, 2006*).

Anisotropy parameters for Cy3-end-labeled BLT and 3'ovr 52-dsRNA (*Figure 6C,G,K*) were similar to the reported values of Cy3-end-labeled dsDNA (*Sanborn et al., 2007*). To obtain precise values for fast correlation times, the fast phase of anisotropy decay was analyzed separately (*Figure 6—figure supplement 1A–J*). Fast ($\phi 1$) and slow ($\phi 2$) rotational correlation times for BLT 52-dsRNA were 0.34 and 2.6ns, respectively, with associated amplitudes of 20% and 80% (*Figure 6K*). For Cy3-end-labeled 52-dsRNA, fast correlation times derive from internal rotational dynamics of the fluorophore, while slow correlation times report on local tumbling of the hydrated nucleic acid (*Sanborn et al., 2007*; *Broos et al., 1995*; *Unruh et al., 2005*).

To gain insights into conformational dynamics of the helicase domain while engaged with BLT termini, we compared anisotropy parameters of BLT 52-dsRNA alone to those obtained with dmDcr-2$^{RIII}$ or dmDcr-2$^{RIII}$•ATPγS (*Figure 6C,D,E,K*). For binding of enzyme (even without nucleotide) to BLT dsRNA, the second phase of the anisotropy decay curve was absent, consistent with a highly restricted local tumbling of hydrated 52-dsRNA inside the helicase domain and/or a marked reduction in the global dynamics of the helicase domain. The fast correlation time became longer by ~3.5-fold (*Figure 6K*), suggesting a reduced conformational fluctuation of the helicase domain in the immediate vicinity of the Cy3 linked to the BLT terminus, which is consistent with the approximately twofold increase in fluorescence life-time of the probe under these conditions (*Figure 1—figure supplement 2E and G*; *Stennett et al., 2015*). Although the fast correlation time for dmDcr-2$^{RIII}$ alone and with ATPγS were similar, the value of $\Delta r$, representing the extent of conformational dynamics, was reduced by ~2.4-fold (*Figure 6D,E,K*), again consistent with an ATPγS-mediated enhanced rigidity in the closed conformational state of the helicase domain (*Sinha et al., 2015*; *Sinha et al., 2018*). In contrast to BLT dsRNA, both internal dynamics and local tumbling of 3'ovr 52-dsRNA were observed when dmDcr-2$^{RIII}$ was added with or without nucleotide (*Figure 6G–K*). Internal dynamics of 3'ovr dsRNA alone (0.32ns) was similar to that observed when dmDcr-2$^{RIII}$ was added (0.35ns), but became approximately twofold slower for dmDcr-2$^{RIII}$ with ATPγS (0.56 ns; *Figure 6K* and *Figure 6—figure supplement 1*), consistent with a smaller change in molecular

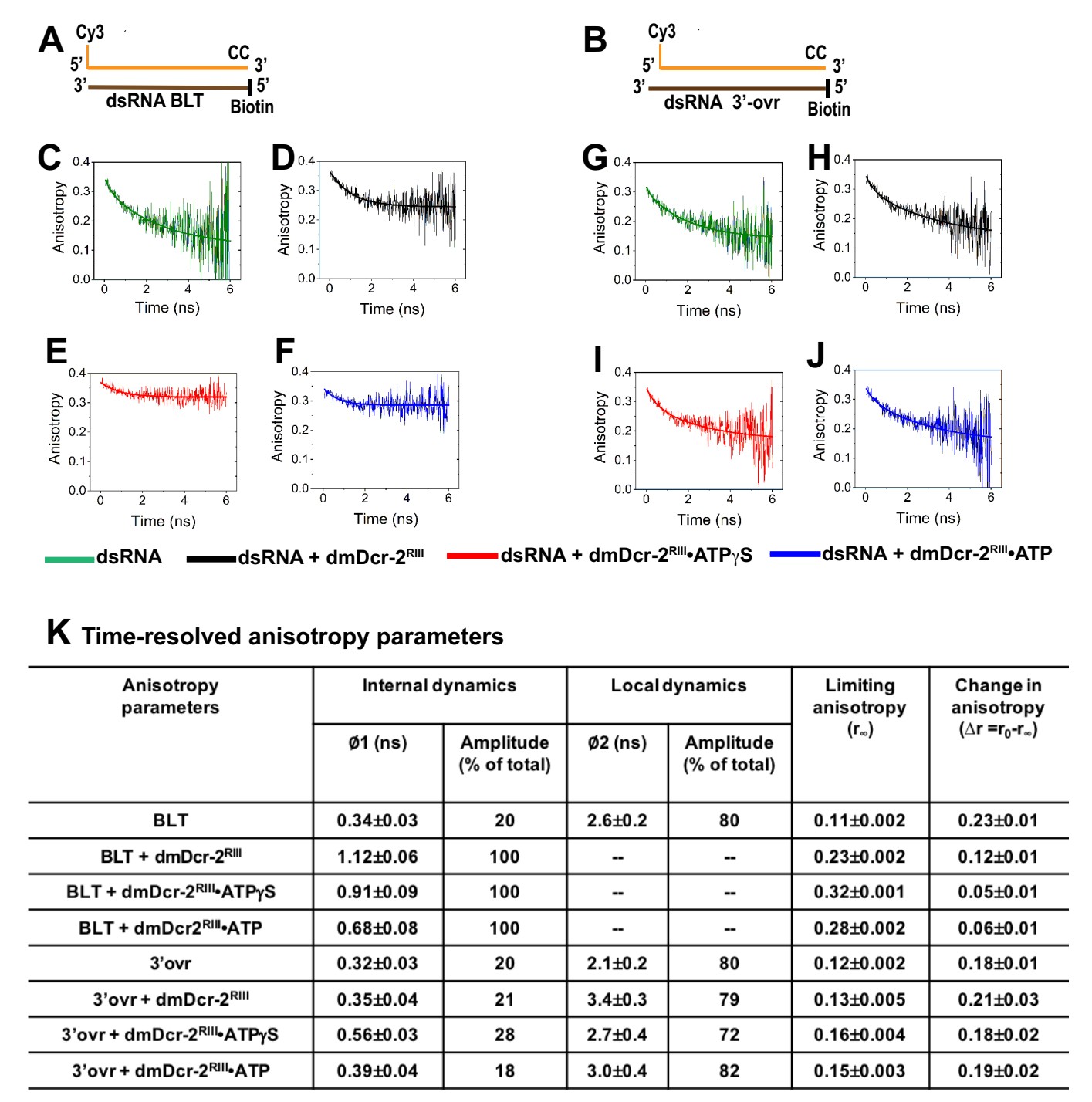

**Figure 6.** Time-resolved fluorescence anisotropy of Cy3-dsRNA alone and bound to the helicase and Platform–PAZ domains of dmDcr-2. Cartoons show Cy3-end-labeled 52-dsRNA with BLT (A) and 2 nt 3'ovr termini (B) with deoxynucleotides (CC) and biotin to prevent dmDcr-2 binding to one end. Representative anisotropy decay curves for Cy3-end-labeled BLT (C–F) and 3'ovr dsRNA (G–J) alone (green) and bound to dmDcr-2[RIII] in the absence of nucleotide (black), or in the presence of ATPγS (red) or ATP (blue). All decay curves were analyzed with a minimum number of exponential terms (single and double exponential rate equations) to obtain the best fit (see Materials and methods), yielding values of anisotropy parameters (K). The precise values of anisotropy parameters associated with the fast correlation time (φ1) were determined from independent analyses of the fast phase of anisotropy decay curves using a single exponential equation (see *Figure 6—figure supplement 1*).

The online version of this article includes the following figure supplement(s) for figure 6:

*Figure 6 continued on next page*

*Figure 6 continued*

**Figure supplement 1.** Fluorescence anisotropy decay curves of Cy3-end-labeled BLT (**A** and **B**) and 3′ovr (**C–J**) 52-dsRNA, −/+ enzyme, and nucleotide as indicated.

**Figure supplement 2.** The fluorescence decay curve of Cy3 attached to 5′-end of BLT (**A**) and 3′ovr (**B**) 52-dsRNA while bound to dmDcr2$^{RIII}$•ATP (**C** and **D**).

rigidity at the 3′ovr terminus compared to BLT when interacting with the helicase domain (*Figure 1— figure supplement 2E–G*). Further, the Δr value for BLT 52-dsRNA bound to dmDcr-2•ATPγS (Δr, 0.05) was ~3.5-fold lower than that for 3′ovr bound to dmDcr-2•ATPγS (Δr, 0.18; *Figure 6K*). This is consistent with the approximately fourfold slower dissociation off-rate of enzyme-bound BLT dsRNA as compared to 3′ovr dsRNA in the presence of ATPγS (*Figure 2*).

To gain insights into how conformational dynamics of the Platform–PAZ domain engaged with dsRNA termini is correlated with substrate cleavage and siRNA release, we compared anisotropy parameters of BLT and 3′ovr 52-dsRNA bound to dmDcr-2$^{RIII}$ with ATP, after translocation places the termini in the Platform–PAZ domain. Notably, the long/slow correlation time ($\phi 2$) was again absent for BLT 52-dsRNA (i.e. with ATP; *Figure 6F,K*), but both fast and slow correlation times were observed for 3′ovr 52-dsRNA in the presence of ATP (*Figure 6J,K*). Furthermore, the fast correlation time was approximately twofold shorter for 3′ovr as compared to BLT termini ($\phi 1$, 0.39ns, 3′ovr; 0.68ns, BLT, *Figure 6K*), consistent with a higher local rigidity of the Platform–PAZ domain at the BLT terminus with ATP (as compared to 3′ovr); this is also consistent with a higher increase in fluorescence life-time of Cy3-BLT dsRNA upon binding to dmDcr-2$^{RIII}$•ATP, which is caused by an increase in molecular rigidity near the fluorophore, as compared to that of 3′ovr (*Figure 6—figure supplement 2*; *Stennett et al., 2015*). Since ATP-dependent kinetic events of the dmDcr-2 catalytic cycle, namely, unwinding, translocation, and dsRNA cleavage are all approximately twofold faster for BLT dsRNA than for 3′ovr dsRNA (*Figures 3E*, *4E,* and *5E*), the reduced fast conformational fluctuation of the Platform–PAZ domain correlates with enzyme catalysis (*Broos et al., 1995*; *Unruh et al., 2005*; *Henzler-Wildman and Kern, 2007*). Additionally, the notable absence of a long correlation time ($\phi 2$) for BLT dsRNA bound to dmDcr-2$^{RIII}$ in the presence of ATP, representing a total loss of local tumbling/rotation of BLT dsRNA within the Platform–PAZ domain prior to its cleavage, suggests that a markedly reduced global dynamics of the Platform–PAZ domain may also contribute to the processive cleavage of BLT dsRNA observed in prior studies (*Welker et al., 2011*; *Sinha et al., 2015*).

## Discussion

Invertebrate Dicers involved in antiviral defense require ATP (*Welker et al., 2011*; *Cenik et al., 2011*; *Sinha et al., 2015*), but how the ATP-dependent helicase domain coordinates Dicer activities, and exactly what these activities are, is poorly understood. Using a real-time, stopped-flow method, and time-resolved fluorescence spectroscopy, we interrogated the role of ATP in dmDcr-2 catalysis, and defined dsRNA binding, unwinding, rewinding, translocation, and cleavage, as reaction intermediates. Our studies provide a kinetic framework for understanding dmDcr-2 as a complex molecular motor (*Figure 7*).

While prior equilibrium studies indicate that 3′ovr dsRNA, but not BLT dsRNA, interacts with dmDcr-2 in the absence of nucleotide, our transient kinetic studies show that dsRNA with either terminus binds in the absence of nucleotide (*Figures 1*, *6,* and *7*). Addition of nucleotide directs both BLT and 3′ovr dsRNA to the helicase domain (*Figures 1* and *2* and *Figure 1—figure supplement 2*), albeit BLT dsRNA is trapped faster and clamped tighter (*Figures 1* and *2*). Prior studies show that dsRNA is unwound by the helicase domain (*Sinha et al., 2018*), and we directly monitored the ATP-dependent transient unwinding and rewinding of dsRNA at the termini (*Figure 3*). Furthermore, we find that the energy of ATP hydrolysis is coupled with the translocation of dmDcr-2 to the cleavage site located ~20–25 nt away from the dsRNA terminus (*Figure 4*). Our studies suggest that ATP binding and hydrolysis by the helicase domain promote terminus-dependent conformational transitions ~80 Å away in the Platform–PAZ domain, and further that these dynamics fine-tune substrate cleavage and siRNA release (*Figures 5* and *6*).

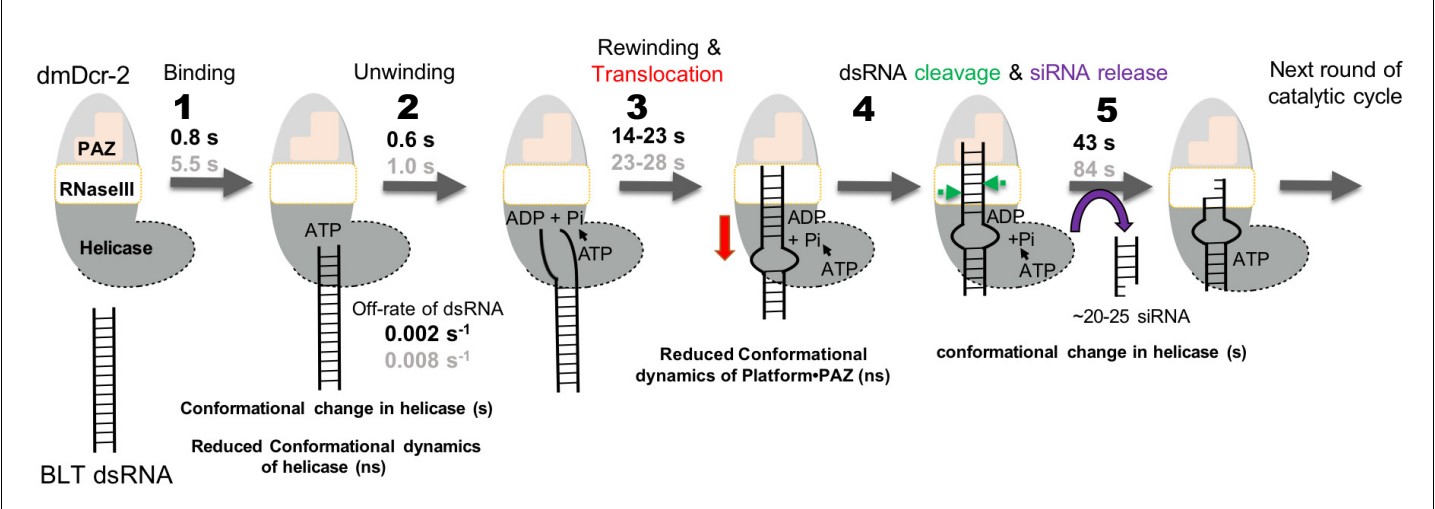

**Figure 7.** Real-time kinetic model for ATP-dependent dmDcr-2-catalyzed reaction with BLT and 3'ovr dsRNA. Kinetic parameters for dmDcr-2 reacting with BLT (black font) and 3'ovr (gray font) dsRNA are reported in seconds (s), and conformational changes noted in seconds or nanoseconds (ns) for numbered steps. (1) ATP binding to the helicase domain promotes a closed conformational state that traps the dsRNA terminus, accompanied by reduced dynamics of the helicase domain; ATPγS allows measurement of off-rates. (2) ATP hydrolysis catalyzes dsRNA unwinding at the terminus. (3) Unwinding is followed by slow rewinding in concert with an ATP-dependent directional translocation of dmDcr-2 (red arrow) to the dsRNA cleavage site (~22 nt from dsRNA terminus). An ATP-dependent modulation in conformational dynamics of the Platform–PAZ domain poises the enzyme for catalysis. Platform–PAZ domains are shown in pink with a shape indicating optimal binding to 3'ovr termini compared to BLT termini. (4) The RNase III domain of dmDcr-2 cleaves (green arrowheads) dsRNA by measuring ~21–22 nts from the terminus bound to the Platform–PAZ domain. (5) The siRNA is released via an ATP-binding mediated conformational change in the helicase domain which shows long range communication with the Platform–PAZ domain bound to siRNA termini.

## Termini-dependent dsRNA discrimination by dmDcr-2 for enhanced antiviral defense

*D. melanogaster* requires the helicase domain of dmDcr-2 to mount an effective antiviral response (*Marques et al., 2013*; *Donelick et al., 2020*), but definitive evidence that this domain discriminates termini of viral dsRNA in vivo, in flies, awaits validation. However, our studies reveal dramatic differences in the interaction and processing of BLT and 3'ovr termini, firmly establishing that the enzyme has evolved the ability to discriminate these termini. Our transient kinetics studies show that dmDcr-2's helicase domain entraps BLT termini approximately sixfold faster, and holds tighter, as compared to 3'ovr termini (*Figures 1* and *2*). Furthermore, a higher observed rate for cleavage of BLT dsRNA ($t_{1/2}$ = 43 s; *Figure 5E*) as compared to its residence time on dmDcr-2, 346 s ($t_{1/2}$ = 0.693/$k_{off}$ *slow*; *Figure 2H*), would be expected to facilitate the *processivity* of dmDcr-2 on long BLT dsRNA observed in prior studies (*Welker et al., 2011*; *Sinha et al., 2015*). In contrast, the residence time of 3'ovr dsRNA on dmDcr-2 in the presence of nucleotide ($t_{1/2}$ = 86 s, *Figure 2H*) is similar to the observed rate of its cleavage ($t_{1/2}$ = 84 s; *Figure 5E*) – indicating the enzyme bound to long 3'ovr dsRNA dissociates at the end of the catalytic cycle after producing a single ~20–25 base-pair siRNA (*distributive cleavage*). While the BLT dsRNA used in our kinetic experiments was designed to mimic termini found on certain viruses, the 3'ovr dsRNA is a product released after the first or subsequent round(s) of the dmDcr-2 catalytic cycle (*Cenik et al., 2011*; *Marques et al., 2013*). Our kinetic studies showing a markedly higher rate of initial engagement and processing of BLT dsRNA (as compared to 3'ovr) by dmDcr-2, underscores the idea that invertebrate Dicers have fine-tuned the transient events of the catalytic cycle (*Benner, 1989*) for an enhanced clearance of newly replicated/intact viral dsRNA (*Flint et al., 2015*; *Mueller et al., 2010*; *Schlee, 2013*; *Sowa et al., 2020*).

## The role of ATP binding and hydrolysis in dmDcr-2 catalysis

Our transient kinetic studies show that ATP binding promotes a termini-dependent initial encounter of dsRNA with dmDcr-2, followed by isomerization of the encounter complex, characterized by a slow conformational change in the helicase domain (*Figure 7*). Using the $t_{1/2}$ values (*Figure 1I*, +

ATPγS), we calculated the observed rate constants ($k_{obs}$ = 0.693/$t_{1/2}$) for the nucleotide-dependent conformational change in dmDcr-2's helicase domain as 0.012 s$^{-1}$ and 0.007 s$^{-1}$ when bound to BLT and 3'ovr dsRNA, respectively. This difference in the kinetics of the nucleotide-mediated conformational change in the helicase domain while bound to distinct dsRNA termini offers a kinetic control mechanism for substrate discrimination by the helicase domain.

We find that ATP-hydrolysis by the helicase domain is essential for energetically costly steps of the catalytic cycle, including unwinding/rewinding and translocation, and conformational dynamics of the Platform–PAZ domain prior to cleavage (*Figure 7*). All of these kinetic events are twofold different for BLT, compared to 3'ovr, dsRNA (*Figures 1*, *3*, *4*, *5*, *6*, and *7*), which correlates well with the twofold difference in ATP-hydrolysis rate for dmDcr-2 bound to these substrates (*Donelick et al., 2020*).

Since the half-lives for ATP-dependent cleavage of BLT and 3'ovr dsRNA, and siRNA release, are the longest steps of the catalytic cycle (*Figure 7*), we assign this as the rate-limiting step. The observed rate constant associated with cleavage/siRNA release with ATP ($t2_{1/2}$, 43.2 s, BLT; 83.8 s, 3'ovr; *Figure 5E*) measured in our FRET-based assay is essentially the same as the rate of isomerization of dmDcr-2•ATPγS•dsRNA ($t2_{1/2}$, 55.5 s, BLT; 95.9 s, 3'ovr, *Figure 1I*), suggesting that the ATP-binding-mediated conformational change in the helicase domain is critical for siRNA release. Such a role for ATP-binding during siRNA release has been proposed for human Dicer (*Zhang et al., 2002*). The regulatory factors Loqs-PD and R2D2 bind dmDcr-2's helicase domain to modulate dsRNA processing and transfer of siRNA to Argonaute-2, respectively (*Liu et al., 2003*; *Liu et al., 2006*; *Marques et al., 2010*; *Hartig and Förstemann, 2011*). We propose that an ATP-mediated conformation change in the helicase domain, in conjunction with R2D2, kinetically fine-tunes the transfer of siRNA from dmDcr-2 to Argonaute-2 at the end of the catalytic cycle.

## The importance of conformational fluctuations of dmDcr-2 in catalysis

Conformational dynamics of proteins are often harnessed by enzymes to enhance catalytic efficiency (*Vendruscolo and Dobson, 2006*; *Boehr et al., 2006*; *Gerstein et al., 1994*; *Henzler-Wildman et al., 2007*; *Benkovic and Hammes-Schiffer, 2003*). Slow transitions in the micro-millisecond range are typically associated with domain movements that facilitate substrate binding and product release (*Gerstein et al., 1994*), while fast, pico-nanosecond fluctuations promote rapid sampling of conformations that can reduce activation barriers and fine-tune transient mechanisms to promote catalysis (*Karplus and Kuriyan, 2005*; *Henzler-Wildman et al., 2007*). Such conformational dynamics are encoded in protein structures, and importantly, a subset of fast-conformational fluctuations (ps–ns) can be kinetically coupled with slower motions (ms–s) to enhance catalytic efficiency (*Karplus and Kuriyan, 2005*; *Gerstein et al., 1994*). Molecular dynamic simulations and NMR studies suggest that conformational fluctuations of proteins are not entirely random, but biased toward specific conformational states that enhance their functional efficiency (*Karplus and Kuriyan, 2005*). This phenomenon is mechanistically analogous to the inherent bias toward selecting a few specific conformational states of a protein, and funneling them to the next step in the protein folding pathway, instead of sampling all possible conformational states (possibly millions) which would require an astronomical amount of time (Levinthal Paradox) (*Karplus and Kuriyan, 2005*).

Our time-resolved fluorescence anisotropy studies uncovered two termini-dependent conformational dynamics, local (φ1) and global (φ2) (*Figure 6K*), for the helicase and Platform–PAZ domains of dmDcr-2. In contrast to results with 3'ovr dsRNA, the φ2 dynamic is completely absent for BLT dsRNA in the presence of nucleotide, indicating entrapment of BLT dsRNA inside the helicase (ATPγS) and Platform–PAZ (ATP) domains (compare *Figure 6E and F* with 6I and J). In other words, BLT dsRNA is unable to freely rotate/tumble because of a marked global rigidification of these domains in the presence of nucleotide.

The approximately twofold difference in termini-dependent rates of ATP-hydrolysis by dsRNA-bound dmDcr-2 observed in prior studies (*Donelick et al., 2020*), and the approximately twofold difference in observed rates of ATP-dependent kinetic events we report here (unwinding, translocation, dsRNA cleavage) correlates with the approximately twofold difference in rate of conformational fluctuation of the Platform–PAZ domain (φ1, 0.39ns, 3'ovr; 0.68ns, BLT, *Figure 6K*), suggesting a role of conformational fluctuations in the nanosecond time scale for dmDcr-2 catalysis. We propose that a markedly slow fluctuation of the helicase and Platform–PAZ domains of dmDcr-2 with BLT dsRNA, in comparison to 3'ovr dsRNA, is correlated with a faster termini-dependent dsRNA binding and siRNA

release. That said, future studies, including mutations in domains of dmDcr-2, will be required to uncover the complete hierarchy of time scales of dmDcr-2 conformational dynamics that are critical for enzyme catalysis (*Henzler-Wildman et al., 2007*).

Our finding that the local tumbling/rotation of BLT 52-dsRNA inside the Platform–PAZ domain with ATP is dramatically restricted (loss of long correlation time, $\phi$2; *Figure 6K*), in contrast to 3'ovr dsRNA, indicates a substrate-induced global rigidification of the PAZ domain that may facilitate the observed ATP-dependent processive cleavage of BLT dsRNA (*Welker et al., 2011*; *Sinha et al., 2015*). Interestingly, the Platform–PAZ domain of dmDcr-2 is physically linked with its RNase III domains through a flexible helix (connector helix) that is conserved among Dicer enzymes (*MacRae et al., 2006*; *Tian et al., 2014*). We propose that a termini-dependent modulation in global dynamics of the Platform–PAZ domain is communicated allosterically via the connector helix to the RNase III active sites to impact rates of substrate cleavage and product release.

# Materials and methods

## Key resources table

| Reagent type (species) or resource | Designation | Source or reference | Identifiers | Additional information |
|---|---|---|---|---|
| Recombinant DNA agent | pFastBac (plasmid) | Thermo Fisher Scientific | Cat# 10360–014 | |
| Cell line (*Spodoptera frugiperda*) | Sf9 cells | Expression System | Cat# 94–001S RID:CVCL_0549 | |
| Strain (*Escherichia coli*) | DH10Bac Competent Cells | Thermo Fisher Scientific | Cat# 10361012 | Chemically competent cells |
| Protein purification reagent | Strep-Tactin | IBA Lifesciences | Cat# 2-1201-010 | |
| RNA labeling reagent | Sulfo-NHS-ester Cyanine3 | Lumiprobe Corporation | Cat# 21320 | |
| RNA labeling reagent | Sulfo-NHS-ester Cyanine5 | Lumiprobe Corporation | Cat# 23020 | |
| RNA synthesis reagent | | Glen Research | http://www.glenresearch.com | Phosphoramidites |
| Nucleotide | ATP | Thermo Fisher | Cat# R1441 | |
| Nucleotide-analog | ATPγS | Sigma | A-1388–25 MG | |
| Software, algorithm | Origin software package | Origin Lab | Origin Lab Corporation | |
| Fast-mixing device | Stopped-flow system | BioLogic Sciences Instruments | SFM 3000 | |
| Optical filter | Long pass- filter | NewPort | 10LWF-550B | |
| TCSPC fluorescence system | Time-correlated single photon counting (TCSPC) module | PicoQuant | PHR 800 and Picoharp 300 | |
| Sequence-based reagent | 52 nt sense strand for BLT and 3' ovr dsRNA | This paper | ssRNA | 5'-GGAGGUAGUAGGUUGUAUAGUAGUAAGACCA-GACCCUAGACCAAUUCAUGCC-3' <u>CC</u> = deoxynucleotide |
| Sequence-based reagent | 52 nt antisense strand for BLT dsRNA | This paper | ssRNA | Biotin-5'-GGCAUGAAUUGGUCUAGGGUCUGGUCUUACUACUAUACAACCUACUACCUCC-3' |
| Sequence-based reagent | 54 nt antisense strand for 3'ovr dsRNA | This paper | ssRNA | Biotin-5-GGCAUGAAUUGGUCUAGGGUCUGGUCUUACUACUAUACAACCUACUACCUCCCC-3' |

*Continued on next page*

*Continued*

| Reagent type (species) or resource | Designation | Source or reference | Identifiers | Additional information |
|---|---|---|---|---|
| Sequence-based reagent | Cy3-end labeled 52 nt sense strand | IDT | ssRNA | 5'Cy3- GGAGGUAGUAGG UUGU AUAGUAGUAAGAC- CAGACCCU AGACCAAUUCA UGCC-3' <u>CC</u> = deoxynucleotide |

## Expression and purification of dmDcr-2 and variants

dmDcr-2, dmDcr-2$^{RIII}$, and dmDcr-2$^{\Delta Hel}$ were expressed and purified using a baculovirus expression system as described (*Sinha and Bass, 2017*). Briefly, recombinant pFastBac plasmid containing the open reading frame for OSF-tagged dmDcr-2 was transformed into DH10Bac competent cells. Recombinant bacmid was isolated and transfected into SF9 cells to make viral stocks (P0, P1, and P2) for protein expression. Recombinant P2 viral stocks were titered and used for large-scale expression. Expressed protein was purified to homogeneity using Strep-Tactin affinity chromatography as described, and purified protein was dialyzed, concentrated, and stored at −80°C in cleavage assay buffer (25 mM Tris pH 8.0, 100 mM KCl, 10 mM MgCl$_2$.6H$_2$O, 1 mM TCEP), supplemented with 20% glycerol.

## RNA preparation

BLT 52-dsRNA was prepared from 52 nt sense and antisense strands, and 3'ovr dsRNA was prepared with 52 nt sense and 54 nt antisense strands (sequences below) as described (*Sinha et al., 2018*); all strands were chemically synthesized using reagents from Glen Research (Sterling VA), and HPLC purified at the DNA/Peptide Synthesis Core facility at the University of Utah. All dsRNAs were prepared with two deoxynucleotides at the 3'-end of the sense strand, and a biotin at the 5'-end of the antisense strand, to facilitate directional binding of dmDcr-2. Single strands were gel purified after PAGE (17% denaturing). dsRNA was prepared by mixing equimolar amounts of sense and antisense RNAs in annealing buffer (50 mM Tris pH 8.0, 20 mM KCl), heating at 95°C for 2 min, and allowing to cool to room temperature for 4 hr. Annealed dsRNA was purified after 8% native PAGE.

> 52 nt sense strand for BLT and 3'ovr dsRNA
> 5'GGAGGUAGUAGGUUGUAUAGUAGUAAGACCAGAC CCUAGACCAAUUCAUG**CC**-3'
> <u>CC</u> = deoxynucleotide
> 52 nt antisense strand for BLT dsRNA
> Biotin-5'- GGCAUGAAUUGGUCUAGGGUCUGGUCUUACUACUAUACAACCUACUACCUCC-3'
> 54 nt antisense strand for 3'ovr dsRNA
> Biotin-5-GGCAUGAAUUGGUCUAGGGUCUGGUCUUACUACUAUACAACCUACUACC UCCCC-3'

## Labeling of RNA

Cy3-5'-end-labeled 52 nt sense RNA (sequence above) was purchased from IDT (Integrated DNA Technologies). For internal labeling of sense and antisense strands, oligonucleotides were synthesized in-house with C6-amine modified uridine at desired locations needed for monitoring duplex unwinding, translocation, and cleavage. The Sulfo-NHS-ester forms of Cyanine dyes (Cy3 or Cy5) were purchased from Lumiprobe (Lumiprobe corporation). C6-amine modified RNA was mixed with 20-fold molar excess of the NHS-ester modified dye in freshly prepared labeling buffer (100 mM sodium tetraborate pH 8.5; pH adjusted with 12.1 M HCl) as described previously (*Joo and Ha, 2012*). The mixture was incubated for 6 hr at room temperature followed by overnight at 4°C. Labeled RNA was ethanol precipitated, rinsed with 70% cold ethanol to remove excess dye, and gel purified after 17% denaturing PAGE prior to annealing to form duplex.

## Transient kinetic experiments

Kinetic events prior to dsRNA cleavage (binding, unwinding/rewinding, and translocation) were investigated separately utilizing dsRNA with fluorophores (Cy3 or Cy3–Cy5) at specific sites, and by monitoring time-dependent changes in fluorophore signal. Transient kinetic analysis of dsRNA binding to dmDcr-2 was performed using a stopped-flow system (SFM 3000, BioLogic Sciences Instruments) under pseudo first-order conditions in cleavage assay buffer (25 mM Tris pH 8.0, 100 mM KCl, 10 mM MgCl$_2$.6H$_2$O, 1 mM TCEP) at 25°C. The premixing concentrations of dmDcr-2 (or dmDcr-2•ATP/ATPγS) and Cy3-end-labeled dsRNA in stopped-flow syringes were 2 µM and 0.2 µM, respectively. The nucleotide-bound form of dmDcr-2 (dmDcr-2•ATP/ATPγS) was prepared by incubating 2 µM dmDcr-2 for 5 min with a large excess (8 mM) of ATP/ATPγS to ensure rapid binding. The binding reaction was started with rapid mixing of equal volumes of enzyme (or enzyme–nucleotide) with dsRNA. Reaction progress was monitored by exciting the sample at 530 nm and measuring fluorescence emission using a 550 nm long pass-filter (NewPort). At least 4–10 kinetic traces were obtained for each experimental condition and values reported as averages. Averaged traces were analyzed using single or double exponential rate equations, yielding values for the observed rate constant ($k_{obs}$ = 0.693/$t_{1/2}$). Data analysis was performed using the Origin software package (OriginLab corporation). A non-ideal (aggregative) behavior of dmDcr-2 at high micromolar concentration precludes an accurate measurement of $k_{obs}$ as a function of enzyme concentration in our experimental conditions.

## Measurement of dissociation off-rates

Dissociation off-rates for enzyme-bound dsRNA were measured by mixing tenfold excess of 52-dsRNA with dmDcr-2•ATPγS•Cy3-dsRNA in stopped-flow syringes; we observed no difference in observed rates of dissociation with further increases in excess dsRNA (15-fold excess). Off-rate measurements were performed in cleavage assay buffer (25 mM Tris pH 8.0, 100 mM KCl, 10 mM MgCl$_2$.6H$_2$O, 1 mM TCEP) at 25°C. Dissociation kinetics were recorded by exciting samples at 530 nm and detecting fluorescence emission using a 550 nm long pass-filter (NewPort). At least 4–10 traces were obtained and values averaged. Averaged traces were analyzed using single or double exponential rate equations to yield dissociation off-rates. Data analysis was with the Origin software package (OriginLab corporation). The activation-free energy was calculated using the Eyring equation (*Equation 1*):

$$\Delta G^{\ddagger} = -\mathrm{RT}\ln(koff\ h/kB\ T) \tag{1}$$

where R is the gas constant (1.986 cal K$^{-1}$ mol$^{-1}$), $T$ is the absolute temperature, $h$ is the Planck's constant (1.58 × 10$^{-34}$ cal s), and $kB$ is the Boltzmann's constant (3.3 × 10$^{-24}$ cal K$^{-1}$).

## Measurement of translocation kinetics

Translocation assays used 52-dsRNA with a Cy3 covalently attached to the 18th nt of the sense strand and were performed in cleavage assay buffer (25 mM Tris pH 8.0, 100 mM KCl, 10 mM MgCl$_2$.6H$_2$O, 1 mM TCEP) at 25°C. Reactions were monitored upon mixing dsRNA with 10-fold excess of dmDcr-2 alone, or bound to nucleotide (dmDcr-2•ATP/ATPγS), in stopped-flow syringes. dmDcr-2•ATP/ATPγS was prepared by incubating 2 µM of dmDcr-2 with 8 mM ATP/ ATPγS for 5 min as described above. The time-dependent arrival of dmDcr-2 near to the Cy3 attached to dsRNA leads to PIFE. At least 4–10 traces were obtained and values averaged. Averaged traces were analyzed using single or double exponential rate equations, yielding values for the observed rate constant ($k_{obs}$ = 0.693/$t_{1/2}$) for the translocation. Data analysis was performed using the Origin software package (OriginLab corporation).

## Measurement of dsRNA cleavage kinetics

Kinetics of dsRNA cleavage and siRNA release were investigated by monitoring the first round of catalytic cycle for ~5 min. The reaction mixture was synchronized at the beginning of the catalytic cycle via rapid mixing using the stopped-flow system, and by using 10-fold excess of dmDcr-2 over dsRNA. The excess enzyme over substrate ensured that dsRNA remained bound during the first turnover, thus making precise measurement of kinetic parameters feasible. dsRNA used in the cleavage assay had a FRET pair at the primary cleavage site, and cleavage and siRNA release were

recorded by exciting at 530 nm (Cy3 excitation) and monitoring the time-dependent change in FRET signal (Cy5 fluorescence emission) using 670 nm long pass filter. Assays were performed in cleavage assay buffer (25 mM Tris pH 8.0, 100 mM KCl, 10 mM MgCl$_2$.6H$_2$O, 1 mM TCEP) at 25°C in the absence and presence of nucleotide (ATP/ATPγS). Nucleotide-bound dmDcr-2 (dmDcr-2•ATP/ATPγS) was prepared by incubating 2 μM of dmDcr-2 with 8 mM ATP/ ATPγS for 5 min. At least 4–10 traces were obtained and values averaged. Averaged traces were analyzed using single or double exponential rate equations, and data analysis was performed using the Origin software package (OriginLab corporation).

## Time-resolved fluorescence experiments
### Experimental setup

> IRF values: The first trial was 40 ps, second trial 60 ps, and so on; average 50 ps.
> Beam polarization: Vertically polarized to interrogate the sample.
> Resolution: 16 ps
> Integration time: 110 s
> Sample description: Condensed phase samples were prepared right before tested and placed into a 4 mm path length cuvette.
> Laser power: 170–175 μW
> Stability: Stability of the beam varied by 400 nW in the first trial and 1 μW in the second trial.
> Wavelength and FWHM: 528.5 nm with a FWHM of 3.5 nm.
> Wavelength ranges for filters and dichroic: 562 short pass dichroic, 20 nm bandpass centered at 586 nm.

### Channel equalization with neutral density

A 1.0 ND Thorlabs filter was added to the vertical channel's collection fiber path to equalize the raw count rate in both polarization channels. Since both data streams are collected simultaneously, this allows for longer integration times that optimize the signal-to-noise ratio in the overall experiment.

## Measurement of fluorescence lifetimes and time-resolved fluorescence anisotropy

Time-resolved photoluminescence experiments on fluorescently labeled RNAs were used to measure fluorescence lifetime and transient fluorescence anisotropy of the Cy3 chromophore. Data were collected with a polarization-resolved epifluorescence setup whose source is the second harmonic of a tunable-wavelength pulsed Ti:sapphire laser (Coherent Chameleon Ultra II, repetition rate of 80 MHz, <200 fs pulse duration). Excitation pulse energy was kept low (2 pJ/pulse) to prevent sample photodamage, and excitation wavelength was set to 529 nm (FWHM ≤3.5 nm). Excitation beam polarization was set to the vertical axis with high extinction ratio polarizers before being directed to the sample with a 562 nm long-pass dichroic, and focused using a 75 mm focal length aspheric lens. Samples were held in a 4 mm pathlength quartz cuvette. Fluorescence was collected with the same lens and filtered with the same dichroic mirror before passing through a bandpass filter (576–596 nm transmission window) to further eliminate any scattered light from the excitation beam. The fluorescence signal was split into vertical and horizontal polarization channels using a set of polarizing beam splitters, and the signal in each channel was detected with identical single-photon Si photodiodes (MPD systems), whose output was routed to a time-correlated single-photon counting (TCSPC) module (PHR 800 and Picoharp 300, PicoQuant). All spectral cleanup, focusing, collection, and detection optical elements were mounted on a 30 mm cage system to prevent deleterious stray light (<1 Hz dark count rates). To match the signal-to-noise ratio in the data of each polarization channel (which have different collection efficiencies), count rates on the TCSPC module were equalized to within a factor of 2 by placing a neutral density filter (optical density = 1.0) in the path of the vertical channel. The instrument response function for setup was ≤60 ps and data resolution was kept at 16 ps. For each sample, data collection was integrated for 110 s. Individual traces of intensity vs. time delay for the vertical and horizontal polarization channels were stored and exported as ASCII files for further processing.

## Data analysis (time-resolved fluorescence)

Raw data for vertically and horizontally polarized fluorescence traces were imported into Matlab for processing and analysis. The temporal axis for each channel was matched to account for different sample-to-detector path lengths (~380 ps shift in time delay axis). Counts due to background signal were estimated as average of signal before the steep rise due to arrival of excitation pulses, and subtracted from data. Quantitative match of the signal collection efficiency in the orthogonal polarization channels was achieved by finding the factor g required to overlap the long-time (> 2 ns) traces recorded for the free chromophore in buffer solution (tail-matching). The resulting fluorescence lifetime (300 ps), anisotropy amplitude (0.40), and rotational diffusion lifetime (440 ps) of this control sample (Cy3-NHS) reproduced previously reported values (*Sanborn et al., 2007*). The scaling factor g for the control sample of each experimental run was used for all samples in that run. This time-shifted, background-subtracted, and scaled data were then used to calculate total fluorescence output of the sample, $I_{tot}(t)$, and the transient fluorescence anisotropy, $r(t)$, with Equations 2 and 3, respectively.

$$I_{tot}(t) = gI_{VV} = 2I_{VH} \tag{2}$$

$$r(t) = \frac{(gI_{VV} - I_{VH})}{(gI_{VV} + 2\,I_{VH})} \tag{3}$$

Total fluorescence output as a function of delay time was described with either a single- or bi-exponential decay using a nonlinear least-squares fitting algorithm whose output was used to calculate the sample's average fluorescence lifetime. Transient anisotropy curves were described with an exponential decay to a constant (typically nonzero) value. To avoid dsRNA cleavage during the longer times required for time-resolved measurements, we used dmDcr-2[RIII], which binds and hydrolyzes ATP similar to the wild-type enzyme (*Cenik et al., 2011*). All incubations were ~5 min so that kinetic events prior to dsRNA cleavage were complete (one catalytic turnover requires 1–2 min; *Figure 5E*). Anisotropy decay curves were analyzed with single and double exponential rate equations yielding rotational correlation time ($\phi$), fractional amplitude ($A$), limiting anisotropy ($r_\infty$), and change in anisotropy ($\Delta r$) (*Figure 6k*). The change in anisotropy ($\Delta r$) was evaluated by subtracting the value of limiting anisotropy ($r_\infty$) from the fundamental/initial anisotropy ($r_0$) (*Lakowicz, 2006*). In order to evaluate precise values of anisotropy parameters associated with sub-nanosecond dynamics, the fast phase of the decay curve was analyzed independently with a single-exponential rate equation. Goodness of fit or precision of fitted parameters was evaluated using the associated *standard error*, *reduced chi-square ($\chi^2$)* values, as well as randomness of residuals.

## Quantification and statistical analysis

At least 4–10 kinetic traces were collected for each experimental condition used in our transient kinetic experiments. They were averaged, and the resulting averaged trace was analyzed using single or double exponential rate equations (*Equations 4 and 5*).

$$RFU = A\,e^{-kobs.t} + offset \tag{4}$$

$$RFU = A1\,e^{-kobs1.t} + A2\,e^{-kobs2.t} + offset \tag{5}$$

where *RFU* is the relative fluorescence signal, *A* is the amplitude associated with the exponential phase, $k_{obs}$ is the observed rate constant, and *offset* is the baseline signal. The relative fluorescence signal was normalized to one as the maximum signal for ease of comparison. The nonlinear least square fitting of the data to the above rate equations was performed using the Levenberg–Marquardt iteration algorithm available in Origin software package (OriginLab corporation). Goodness of fit or precision of fitted parameters was evaluated using the associated *standard error*, *reduced chi-square ($\chi^2$)* values, and randomness of residuals.

## Acknowledgements

We thank the members of the Bass lab for helpful discussions. This work was supported by funds to BLB from the National Institute of General Medical Sciences (R01GM121706; R35GM141262) and startup funds to RN from the University of Utah Chemistry Department.

## Additional information

### Funding

| Funder | Grant reference number | Author |
|---|---|---|
| National Institute of General Medical Sciences | R01GM121706 | Brenda L Bass |
| University of Utah | Startup funds | Rodrigo Noriega |
| National Institute of General Medical Sciences | R35GM141262 | Brenda L Bass |

The funders had no role in study design, data collection and interpretation, or the decision to submit the work for publication.

### Author contributions

Raushan K Singh, Conceptualization, Data curation, Software, Formal analysis, Validation, Investigation, Visualization, Methodology, Writing - original draft, Writing - review and editing; McKenzie Jonely, Data curation, Software, Formal analysis, Investigation, Methodology; Evan Leslie, Nick A Rejali, Methodology; Rodrigo Noriega, Resources, Data curation, Software, Formal analysis, Funding acquisition, Methodology, Writing - review and editing; Brenda L Bass, Conceptualization, Resources, Supervision, Funding acquisition, Validation, Investigation, Writing - original draft, Project administration, Writing - review and editing

### Author ORCIDs

Raushan K Singh (ID) https://orcid.org/0000-0002-3636-9112
Nick A Rejali (ID) http://orcid.org/0000-0001-7210-9425
Brenda L Bass (ID) https://orcid.org/0000-0003-1728-2254

### Decision letter and Author response

Decision letter https://doi.org/10.7554/eLife.65810.sa1
Author response https://doi.org/10.7554/eLife.65810.sa2

## Additional files

### Supplementary files

• Transparent reporting form

### Data availability

All data generated or analysed during this study are included in the manuscript and supporting files.

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
