## [Decision Letter]

**Acceptance summary:**

The present paper provides a kinetic analysis of a Dicer-family enzyme, *Drosophila* Dicer-2, that is of interest to scientists within the field of RNA editing and gene expression. The findings provide a better understanding of the role that the helicase domain of the protein plays in coupling ATP turnover to RNA end discrimination and RNA editing. The findings inform our understanding Dicer mechanism and its ability to distinguish between self and non-self RNAs. The impact of the study extends to an audience beyond the immediate RNA community, because of the pivotal role of Dicer and related enzymes in cellular function and in immunity, and because there is a dearth of rigorous, quantitative biophysical studies on proteins that interact with RNA. Overall, the work adds new, important insight to our mechanistic understanding of RNA-protein interactions.

**Decision letter after peer review:**

Thank you for submitting your article "Transient kinetic studies of the antiviral *Drosophila* Dicer-2 reveal roles of ATP in self–nonself discrimination" for consideration by *eLife*. Your article has been reviewed by 2 peer reviewers, and the evaluation has been overseen by a Reviewing Editor and Michael Marletta as the Senior Editor. The reviewers have opted to remain anonymous.

Reviewer #2 (Recommendations for the authors):

1. It was difficult to follow the evidence that suggests that the 3' ovr dsRNA does not bind to the PAZ when ATP is present or that the helicase domain loses its specificity and binds to both types of RNAs (meaning that its affinity to 3'ovr increases with ATP relative to the affinity of PAZ for 3'ovr). If Kd values are measured previously to support this claim they should be discussed. The disappearance of one phase upon deletion of Hel is not a strong evidence, because ATP effects cannot be tested with this mutant. An RNA binding mutant of the helicase domain would be more appropriate here.

2. Are the measured off rates and on rates of the two RNAs consistent with the reported Kd values of these RNA complexes?

3. To support the FRET unwinding experiments, it is important that the authors show both acceptor fluorescence increase (shown) and donor fluorescence decrease (not shown). The mirroring of the signals will be a more definitive proof for a FRET signal supporting transient dsRNA unwinding while distinguishing any signals from protein binding and translocation.

Reviewer #3 (Recommendations for the authors):

1. In my estimate the model should be updated to include the effects of ATP, ATPγS and the different overhangs on the kinetic parameters.

2. This reviewer is skeptical about the value of the anisotropy decay measurements for the study. This would require substantial additional experimentation at different conditions (to alter the fluctuations) and, ideally with enzyme mutation (to allow for mechanistic interpretations), for comparably small benefits for the current manuscript. If the authors wish to retain this section in the manuscript, a more detailed discussion of caveats associated with the current interpretation is, however, essential. Another option is to eliminate or otherwise de-emphasize this part.

3. More information on the binding step needs to be provided.

4. Provide control that excess of unlabeled RNA to trap dissociating enzyme in actually traps all released (and otherwise present) protein (Figure 2, point 6).

5. Provide kinetic data from PAGE timecourse(s) for substrate cleavage, to validate the model. Those data probably already exist with the substrates at the conditions tested.

6. The authors suggest that a higher observed rate for cleavage of BLT dsRNA, compared to its residence time on dmDcr-2 enhances the processivity of dmDcr-2 on long BLT dsRNA. In the context of the study, processivity (multiple consecutive translocation steps) has neither been demonstrated nor has it been measured. Revise this passage.

---

## [Author Response]

Reviewer #2 (Recommendations for the authors):1. It was difficult to follow the evidence that suggests that the 3' ovr dsRNA does not bind to the PAZ when ATP is present or that the helicase domain loses its specificity and binds to both types of RNAs (meaning that its affinity to 3'ovr increases with ATP relative to the affinity of PAZ for 3'ovr). If Kd values are measured previously to support this claim they should be discussed. The disappearance of one phase upon deletion of Hel is not a strong evidence, because ATP effects cannot be tested with this mutant. An RNA binding mutant of the helicase domain would be more appropriate here.

We agree with this reviewer that the dsRNA binding experiments of Figure 1, while supporting a switch in binding sites, do not by themselves substantiate the existence of such a switch. Only by the iterative analysis of multiple kinetic experiments to monitor binding, unwinding and translocation, that are reported in our manuscript, is this conclusion validated. Evidence for the binding site switch is given by:

a. Figure 1 (e.g., Figure 1I) shows that the kinetics for dmDcr-2 binding to a 3’ovr dsRNA are monophasic in the absence of nucleotide, but become biphasic with nucleotide. The second-order rate constant for the bimolecular step with ATPγS (1.6×10^5^ M^-1^ s^-1^, i.e. *k_^+^1_*, see Figure 2-supplement 1) is an order of magnitude higher than that without nucleotide (2.5×10^4^ M^-1^ s^-1^; in the single step binding of 3’ovr dsRNA to enzyme without nucleotide (Figure 1D, black trace) *k_^+^1_* = *k_obs_*/[dmDcr-2] when *k_^+^1_*>>*k_-1_*). In considering the data reported in our manuscript, it seems very unlikely that the change from monophasic to biphasic kinetics, and the order of magnitude change in rate constant, could represent anything other than a change in binding site. Again, however, we emphasize that this conclusion takes into account all of the experiments reported in our manuscript.

b. The ATP-dependent unwinding of BLT dsRNA by dmDcr-2 occurs in the helicase domain, as first visualized in our cryo-EM structure (Sinha et al., 2018). In our manuscript we monitor unwinding/rewinding (Figure 3), and indeed, unwinding is completely dependent on ATP, and importantly, is observed with both BLT and 3’ovr dsRNA. While unwinding of 3’ovr dsRNA is 2-fold slower than that of BLT dsRNA, the similarity of the measured kinetic parameters (Figure 3, see panel E) indicates that both BLT and 3’ovr dsRNA are binding and being unwound by the helicase domain. By contrast, while unwinding does not occur without nucleotide, dmDcr-2 mediates slow cleavage of 3’ovr dsRNA in the absence of nucleotide (Figure 5D, black trace), consistent with the nucleotide-independent binding of this dsRNA terminus to the Platform–PAZ domain. Further, prior steady state assays and our kinetic studies have demonstrated that this nucleotide-independent cleavage (~20-23 nt from the dsRNA terminus) occurs even with a truncated protein lacking the helicase domain (Figure 5—supplement 2C and D, cyan trace). In order to emphasize the latter point, we have extended a sentence to acknowledge this prior observation (Page 14, line 308; sentence beginning “This result is consistent with prior studies…”).

c. The presence of a helicase-mediated and ATP-dependent arrival/translocation of dmDcr-2 to the enzyme’s cleavage site on 3’ovr dsRNA (prior to substrate cleavage; Figure 5D, blue trace), in contrast to the complete lack of translocation without ATP (Figure 5D, black trace), again supports a nucleotide-dependent switch in 3’ovr dsRNA binding site from the Platform–PAZ domain to the helicase domain.

In regard to other aspects of this comment: (1). We agree that the ∆Hel mutant protein cannot be used to monitor ATP-dependent effects, and note that the included ∆Hel mutant experiments were performed only to interrogate binding in the absence of nucleotide. To clarify this further, we have added the phrase “..in the absence of nucleotide..” to the sentence on page 5, line 104; sentence beginning “Indeed, experiments performed with dmDcr-2…”. (2). Kd values that separately interrogate interactions with the Platform–PAZ domain versus the helicase domain are not available, unfortunately. We also note that in our transient kinetic studies, the initial encounter of a dsRNA terminus (with or without nucleotide) to distinct domains of dmDcr-2 is primarily dictated by its binding *on-rates*, unlike steady-state assays where equilibrium binding affinity (Kd) determines ligand binding specificity.

2. Are the measured off rates and on rates of the two RNAs consistent with the reported Kd values of these RNA complexes?

We considered this a very important issue to clarify and have added a new Supplemental figure that includes detailed information for evaluating Kd values for dsRNA binding to dmDcr-2•ATPγS, using the microscopic rate constants obtained from our transient studies (Figure 2—Supplement 1 of revised manuscript). As illustrated in the new Supplemental figure, the Kd values estimated from the *on*- and *off-rates* are in agreement with those of reported Kd values obtained from equilibrium binding assays. Here we note that small differences in the kinetically- and equilibrium-determined Kd values for dsRNA interaction with dmDcr-2 could arise due to differences in methods and/or the experimental conditions used in these assays, such as, protein/dsRNA concentration and whether reactions were performed with ATP or a non-hydrolyzable analog (Jarmoskaite et al., 2020).

3. To support the FRET unwinding experiments, it is important that the authors show both acceptor fluorescence increase (shown) and donor fluorescence decrease (not shown). The mirroring of the signals will be a more definitive proof for a FRET signal supporting transient dsRNA unwinding while distinguishing any signals from protein binding and translocation.

We thank the reviewer for appreciating our experimental strategy, and we agree that it would be ideal to show both donor and acceptor fluorescence. However, for reasons described below, this approach is problematic. Notably, dsRNA substrates used for binding and unwinding/rewinding studies (Figures 1 and 3) have Cy3 at the same site within the dsRNA. During the fast phase of dsRNA binding to dmDcr-2 with nucleotide, the donor (Cy3) fluorescence is enhanced upon experiencing a rigid microenvironment inside the helicase domain (Figure 1 and Figure 1—supplement 2). Importantly, the time-scales for the fast binding and transient unwinding phase of dsRNA termini are similar: For BLT, compare Figure 1I, +ATP, 0.8s and Figure 3E +ATP, 0.5s; for 3’ovr, compare Figure 1I, +ATP, 5.5s and Figure 3E +ATP, 1s. Thus, the Cy3 signal from binding would interfere with any enhancement of the Cy3 signal from a distance change in donor-acceptor pair during transient unwinding. In order to mitigate such interference, we chose to monitor the Cy5 FRET signal to monitor the ATP-dependent transient unwinding of dsRNA catalyzed by dmDcr-2. (We note that in the translocation experiments of Figure 4, the half-lives for translocation to the Cy3-labeled 18^th^ base pair of the dsRNA are 14s (BLT) and 29s (3’ovr), and therefore, they are kinetically well-separated from the transient unwinding events (*t_1/2_*, 0.5-1.0s) thus precluding any interference.)

To address this reviewer’s concern, we have added a sentence describing our rationale in the revised manuscript (Page 10, Lines 203-208; sentence beginning, “Instead of donor (Cy3) fluorescence…”).

Reviewer #3 (Recommendations for the authors):1. In my estimate the model should be updated to include the effects of ATP, ATPγs and the different overhangs on the kinetic parameters.

We agree with this reviewer and have modified our model shown in Figure 7 to more fully illustrate the data reported in our paper. In the modified figure we incorporate information on the impact of ATP and ATPγS, and provide kinetic parameters that allow comparisons of blunt and 3’ overhanging termini. We have also revised the legend to clarify information in our revised figure.

2. This reviewer is skeptical about the value of the anisotropy decay measurements for the study. This would require substantial additional experimentation at different conditions (to alter the fluctuations) and, ideally with enzyme mutation (to allow for mechanistic interpretations), for comparably small benefits for the current manuscript. If the authors wish to retain this section in the manuscript, a more detailed discussion of caveats associated with the current interpretation is, however, essential. Another option is to eliminate or otherwise de-emphasize this part.

We agree with this reviewer that a complete understanding of the observed correlation will require additional experimentation that is beyond the scope of this manuscript, and we have added a sentence to our revised manuscript to address this issue (Page 22, Lines 521-523, sentence beginning, “That said, future studies,…”). We have also expanded our discussion of how conformational fluctuations can be coupled to slower motions to enhance catalysis (Page 21, Lines 495-505; second half of first paragraph of section called “The importance of conformational fluctuations…”).

We do feel strongly that the results of our fluorescence anisotropy experiments are exciting and set the stage for further experimentation, and we have retained these experiments in our revised manuscript. However, this reviewer’s comments made us realize that our description of these experiments was confusing and their logic was sometimes lost. To address this issue, we have made small changes throughout the Result section describing these experiments (Page 15, line 319; section called “Conformational dynamics of helicase and Platform–PAZ domains impact dmDcr-2 catalysis”). Similarly, we have revised the section of the Discussion relating to these experiments (Page 21-23, lines 487-557; section called “The importance of conformational fluctuations of dmDcr-2 in catalysis”).

3. More information on the binding step needs to be provided.

We have studied the second-order reaction of dsRNA binding to dmDcr-2 under pseudo first-order conditions, where the concentration of enzyme is so large (compared to substrate) that it is regarded as constant during the binding reaction. The pre-mixing concentration of dmDcr-2 (2μM) is 10-fold-excess over dsRNA (0.2μM) in the stopped-flow syringes. The equilibrium binding affinities of BLT and 3’ovr dsRNA to dmDcr-2 are ~50-60nM (Donelick et al., 2020), and therefore, the enzyme is saturating with respect to the substrate in our experimental condition. However, we note that the precise measurement of the second-order rate constant is not contingent upon the use of a saturating concentration of dmDcr-2 if the pseudo-first condition is met. The concentrations of dsRNA and dmDcr-2 were carefully chosen so that there is substantial change in the fluorescence signal (with a high signal to noise ratio) arising from the change in free Cy3-dsRNA concentration upon its binding to dmDcr-2 over time, and to avoid any kinetic complexity linked with potential non-ideal (aggregative) behavior of dmDcr-2 at high μM concentration. We have estimated the second-order rate constant (*k_^+^1_*) for binding of dsRNA using the relevant kinetic scheme/equation, and the concentration of enzyme after mixing (1µM; see new Figure 2—supplement 1A-C), and found values to be 1.1×10^6^ M^-1^s^-1^ and 1.6×10^5^ M^-1^s^-1^, respectively, for BLT and 3’ovr dsRNA binding to dmDcr-2•ATPγS. The kinetically-determined Kd values for the enzyme-substrate interaction in the presence of ATPγS is in close agreement with Kd values obtained from equilibrium binding studies (Figure 2—supplement 1 of revised manuscript)—suggesting the validity of our *minimal* kinetic model without an additional isomerization step.

The half-life for the fast phase of BLT dsRNA binding to dmDcr-2 with ATP (0.8±0.2s) and ATPγS (0.6±0.1s) are essentially the same within the limit of experimental error (Figure 1I). However, this reviewer’s comment made us realize that using 0.6 is confusing. All of the nucleotide-dependent steps in Figure 7 have now been changed to the ATP values.

4. Provide control that excess of unlabeled RNA to trap dissociating enzyme in actually traps all released (and otherwise present) protein (Figure 2, point 6).

In our dissociation off-rate measurements, a further 5-fold increase in the concentration of unlabeled dsRNA (total of 15-fold excess) did not alter the observed rate of dissociation. We thank the reviewer for realizing we had neglected to mention this. We have added a line to the methods noting this important control (Page 35, lines 807-808 in revised manuscript; first sentence of section called, “Measurement of dissociation off-rates”).

5. Provide kinetic data from PAGE timecourse(s) for substrate cleavage, to validate the model. Those data probably already exist with the substrates at the conditions tested.

Although gel-based PAGE assays allow easy visualization of the stable intermediates and/or end products of an enzyme-catalyzed reaction, they do not provide any information about the transient intermediates that are generated and decayed in the millisecond-second time-scale of the catalytic cycle. This is primarily because PAGE assays rely on manual mixing, which is inefficient, and takes several seconds. Since one turnover of the dmDcr-2 catalytic cycle is completed within 43-84 seconds (e.g. see Figure 5E), a reliable interrogation of the transient kinetic mechanism of dmDcr-2 without employing a rapid/fast-mixing device (stopped-flow system) is infeasible. In our stopped-flow system, homogenous mixing of dsRNA with dmDcr-2 is completed within 1.2 milliseconds (dead time). Due to the fast mixing, we were able to dissect the sequence of key molecular events of the dmDcr-2 catalytic cycle: binding, unwinding/rewinding, translocation, and dsRNA cleavage/siRNA release. Nonetheless, the kinetic parameters we report here are consistent with observations made in our prior end-point assays. For example, the higher rate of ATP-dependent cleavage of BLT dsRNA, as compared to 3’ovr, measured in our transient kinetic studies, are qualitatively similar to the termini-dependent cleavage rates observed with dmDcr-2 by our lab previously, using a PAGE assay (Sinha et al., 2015).

6. The authors suggest that a higher observed rate for cleavage of BLT dsRNA, compared to its residence time on dmDcr-2 enhances the processivity of dmDcr-2 on long BLT dsRNA. In the context of the study, processivity (multiple consecutive translocation steps) has neither been demonstrated nor has it been measured. Revise this passage.

While prior equilibrium studies from our laboratory (Welker et al. MC 2011; Sinha et al., MC 2015) indicate dmDcr-2 acts processively with blunt dsRNA and ATP, we completely agree that the transient kinetic studies we describe here do not measure processivity, but rather, a single catalytic cycle of dmDcr-2 that produces one siRNA. Nonetheless, our experiments indicate that the enzyme remains bound to BLT dsRNA after producing one siRNA, in contrast to 3’ovr dsRNA where dmDcr-2 dissociates at the end of the catalytic cycle; the latter observations are consistent with the processivity observed for BLT, but not 3’ovr, dsRNA in prior studies.

In light of this reviewer’s comment we have revised the noted sentence (Page 19, paragraph 1, lines 435-439; sentence beginning, “Furthermore, a higher observed rate for cleavage of BLT dsRNA…”) to clarify that we are discussing how the experiments reported here fit with the previously reported processivity. In revising our manuscript, we have paid attention to this issue at other times where processivity is mentioned (Page 17, lines 394-405; sentence beginning, “Additionally, the notable absence of a long correlation time…”; Page 22-23, lines 524-552; last paragraph of Discussion, “Our finding that…”).